# SIGN BITS ARE ALL YOU NEED FOR BLACK-BOX ATTACKS

**Abdullah Al-Dujaili**
CSAIL, MIT
Cambridge, MA 02139
aldujail@mit.edu

**Una-May O'Reilly**
CSAIL, MIT
Cambridge, MA 02139
unamay@csail.mit.edu

## ABSTRACT

We present a novel black-box adversarial attack algorithm with state-of-the-art model evasion rates for query efficiency under $\ell_\infty$ and $\ell_2$ metrics. It exploits a *sign-based*, rather than magnitude-based, gradient estimation approach that shifts the gradient estimation from continuous to binary black-box optimization. It adaptively constructs queries to estimate the gradient, one query relying upon the previous, rather than re-estimating the gradient each step with random query construction. Its reliance on sign bits yields a smaller memory footprint and it requires neither hyperparameter tuning or dimensionality reduction. Further, its theoretical performance is guaranteed and it can characterize adversarial subspaces better than white-box gradient-aligned subspaces. On two public black-box attack challenges and a model robustly trained against transfer attacks, the algorithm's evasion rates surpass all submitted attacks. For a suite of published models, the algorithm is $3.8\times$ less failure-prone while spending $2.5\times$ fewer queries versus the best combination of state of art algorithms. For example, it evades a standard MNIST model using just 12 queries on average. Similar performance is observed on a standard IMAGENET model with an average of 579 queries.

## 1 INTRODUCTION

***Problem.*** Deep Neural Networks (DNNs) are vulnerable to adversarial examples, which are malicious inputs designed to fool the model's prediction—see (Biggio and Roli, 2018) for a comprehensive, recent overview of adversarial examples. Research on generating these malicious inputs started in the *white-box* setting, where access to the gradients of the models is assumed. Since the gradient points to the direction of steepest ascent, an input can be perturbed along the gradient's direction to maximize the network's loss, thereby potentially causing misclassification under class prediction, e.g. with images, or evasion under detection, e.g. with malware. The assumption of access to the underlying gradient does not however reflect real world scenarios. Attack algorithms under a more realistic, restrictive *black-box* threat model, which assumes access to predictions in lieu of gradients, are therefore studied. Central to their approaches is estimating the gradient. To estimate the magnitudes and signs of the gradient, the community at large has formulated a continuous optimization problem of $O(n)$ complexity where $n$ is the input dimensionality. Most recently work has sought to reduce this complexity by means of data-/time-dependent priors Ilyas et al. (2019). In this paper, we take a different tact and reduce the central problem to just estimating the signs of the gradients. Our intuition arises from observing that estimating the sign of the top 30% gradient coordinates by magnitude is enough to achieve a rough misclassification rate of 70%. Figure 1 reproducing Ilyas et al. (2019) illustrates this observation for the MNIST dataset–see Appendix A for other datasets. Therefore our goal is to recover the sign of the gradient with high query efficiency so we can use it to generate adversarial examples as effective as those generated by full gradient estimation approaches.

***Related Work.*** We organize the related work in two themes, namely *Adversarial Example Generation* and *Sign-Based Optimization*. The literature of the first theme primarily divides into *white-box* and *black-box* settings. The white-box setting, while not the focus of this work, follows from the works of Biggio et al. (2013) and Goodfellow et al. (2015) who introduced the Fast Gradient Sign Method (FGSM), including several methods to produce adversarial examples for various learning tasks and threat perturbation constraints (Carlini and Wagner, 2017; Moosavi-Dezfooli et al., 2016; Hayes and

Danezis, 2017; Al-Dujaili et al., 2018; Kurakin et al., 2017; Shamir et al., 2019). Turning to the *black-box* setting and iterative optimization schemes, Narodytska and Kasiviswanathan (2017), without using any gradient information, use a naive policy of perturbing random segments of an image to generate adversarial examples. Bhagoji et al. (2017) reduce the dimensions of the feature space using Principal Component Analysis (PCA) and random feature grouping, before estimating gradients. Chen et al. (2017) introduce a principled approach by using gradient based optimization. They employ finite differences, a zeroth-order optimization means, to estimate the gradient and then use it to design a gradient-based attack. While this approach successfully generates adversarial examples, it is expensive in how many times the model is queried. Ilyas et al. (2018) substitute traditional finite differences methods with Natural Evolutionary Strategies (NES) to obtain an estimate of the gradient. Tu et al. (2018) provide an adaptive random gradient estimation algorithm that balances query counts and distortion, and introduces a trained auto-encoder to achieve attack acceleration. Ilyas et al. (2019) extend this line of work by proposing the idea of gradient priors and bandits: $\mathtt{Bandits}_{TD}$. Our work contrasts with the general approach of these works in two ways: a) We focus on estimating the *sign* of the gradient and investigate whether this estimation suffices to efficiently generate adversarial examples. b) The above methods employ random sampling in constructing queries to the model while our construction is *adaptive*.[1] Another approach involves learning adversarial examples for one model (with access to its gradient information) to transfer them against another (Liu et al., 2016; Papernot et al., 2017). Alternately, Xiao et al. (2018) use a Generative Adversarial Network (GAN) to generate adversarial examples which are based on small norm-bounded perturbations. These methods involve learning on a different model, which is expensive, and not amenable to comparison with setups—including ours—that directly query the model of interest.

*Sign-Based Optimization.* In the context of general-purpose continuous optimization methods, sign-based stochastic gradient descent was studied in both zeroth- and first-order setups. In the latter, Bernstein et al. (2018) analyzed $\mathtt{signSGD}$, a sign-based $\mathtt{Stochastic\ Gradient\ Descent}$, and showed that it enjoys a faster empirical convergence than $\mathtt{SGD}$ in addition to the cost reduction of communicating gradients across multiple workers. Liu et al. (2019) extended $\mathtt{signSGD}$ to zeroth-order setup with the $\mathtt{ZO\text{-}SignSGD}$ algorithm. $\mathtt{ZO\text{-}SignSGD}$ (Liu et al., 2019) was shown to outperform $\mathtt{NES}$ against a black-box model on MNIST. These approaches *use* the sign of the gradient (or its zero-order estimate) to achieve better convergence, whereas our approach both *estimates* and *uses* the sign of the gradient.

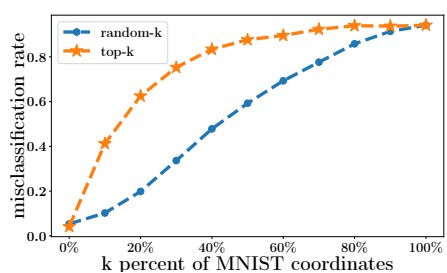

Figure 1: Misclassification rate of an MNIST model on the *noisy* $\mathtt{FGSM}$'s adversarial examples as a function of correctly estimated coordinates of $\text{sign}(\nabla_{\boldsymbol{x}} f(\boldsymbol{x}, y))$ on 1000 random MNIST images. Estimating the sign of the top 30% gradient coordinates (in terms of their magnitudes) is enough to achieve a rough misclassification rate of 70%. More details can be found in Appendix A.

***Contributions.*** We present the following contributions at the intersection of adversarial machine learning and black-box (zeroth-order) optimization: 1) We exploit the separability property of the directional derivative of the loss function of the model under attack in the direction of $\{\pm 1\}^n$ vectors, to propose a *divide-and-conquer*, *adaptive*, *memory-efficient* algorithm, we name $\mathtt{SignHunter}$, to estimate the gradient sign bits. 2) We provide a worst-case theoretical guarantee on the number of queries required by $\mathtt{SignHunter}$ to perform at least as well as $\mathtt{FGSM}$ (Goodfellow et al., 2015), which has access to the model's gradient. To our knowledge, no black-box attack from the literature offers a similar performance guarantee. 3) We evaluate our approach on a rigorous set of experiments on both, standard and adversarially hardened models. All other previous works on this topic have published their results on a subset of the datasets and threat models we experimentally validate in this work. Through these experiments, we demonstrate that $\mathtt{SignHunter}$'s adaptive search for the gradient sign allows it to craft adversarial examples within a mere fraction of the theoretical number of queries thus outperforming $\mathtt{FGSM}$ and state-of-the-art black-box attacks. 4) We release a software framework to systematically benchmark adversarial

---

[1]We use the term adaptive here to characterize how the algorithm constructs its perturbation vector $\boldsymbol{q}_t$ deterministically based on the previous perturbations. Mathematically, the perturbation vector at time $t$ can be expressed as deterministic function $\boldsymbol{q}_t = g(\boldsymbol{q}_{t-1})$. In contrast, other algorithms construct I.I.D randomly perturbation vectors: $\boldsymbol{q}_t \sim \mathcal{N}(0, I)$.

black-box attacks, including `SignHunter`'s, on MNIST, CIFAR10, and IMAGENET models in terms of success rate, query count, and other metrics. 5) We demonstrate how `SignHunter` can be used to characterize adversarial cones in a black-box setup and in doing so, highlight the gradient masking effect.

***Notation.*** Let $n$ denote the dimension of datapoint $\boldsymbol{x}$. Denote a hidden $n$-dimensional binary code by $\boldsymbol{q}^*$. That is, $\boldsymbol{q}^* \in \mathcal{H} \equiv \{-1, +1\}^n$. Further, denote the directional derivative of some function $f$ at a point $\boldsymbol{x}$ in the direction of a vector $\boldsymbol{v}$ by $D_{\boldsymbol{v}} f(\boldsymbol{x}) \equiv \boldsymbol{v}^T \nabla_{\boldsymbol{x}} f(\boldsymbol{x})$ which often can be approximated by the *finite difference* method. That is, for $\delta > 0$, we have

$$D_{\boldsymbol{v}} f(\boldsymbol{x}) = \boldsymbol{v}^T \nabla_{\boldsymbol{x}} f(\boldsymbol{x}) \approx \frac{f(\boldsymbol{x} + \delta \boldsymbol{v}) - f(\boldsymbol{x})}{\delta} \ . \tag{1}$$

Let $\Pi_S(\cdot)$ be the projection operator onto the set $S$, $B_p(\boldsymbol{x}, \epsilon)$ be the $\ell_p$ ball of radius $\epsilon$ around $\boldsymbol{x}$.

## 2    GRADIENT ESTIMATION

At the heart of black-box adversarial attacks is generating a *perturbation vector* to slightly modify the original input $\boldsymbol{x}$ so as to fool the network prediction of its true label $y$. Put differently, an adversarial example $\boldsymbol{x}'$ maximizes the network's loss $L(\boldsymbol{x}', y)$ but still remains $\epsilon$-close to the original input $\boldsymbol{x}$. Although the loss function $L$ can be non-concave, gradient-based techniques are often very successful in crafting an adversarial example Madry et al. (2017). That is, setting the perturbation vector as a step in the direction of $\nabla_{\boldsymbol{x}} L(\boldsymbol{x}, y)$. Consequently, the bulk of black-box attack methods try to *estimate the gradient* by querying an oracle that returns, for a given input/label pair $(\boldsymbol{x}, y)$, the value of the network's loss $L(\boldsymbol{x}, y)$, consulting prediction or classification accuracy. Using only such value queries, the basic approach relies on the *finite difference method* to approximate the directional derivative (Eq. 1) of the function $L$ at the input/label pair $(\boldsymbol{x}, y)$ in the direction of a vector $\boldsymbol{v}$, which corresponds to $\boldsymbol{v}^T \nabla_{\boldsymbol{x}} L(\boldsymbol{x}, y)$. With $n$ linearly independent vectors $\{\boldsymbol{v}_i^T \nabla_{\boldsymbol{x}} L(\boldsymbol{x}, y) = d_i\}_{1 \leq i \leq n}$, one can construct a linear system of equations to recover the full gradient. Clearly, this approach's query complexity is $O(n)$, which can be prohibitively expensive for large $n$ (e.g., $n = 268,203$ for the IMAGENET dataset). Recent works try to mitigate this issue by exploiting data- and/or time-dependent priors (Tu et al., 2018; Ilyas et al., 2018; 2019). However, the queries are not adaptive, they are constructed based on i.i.d. random vectors $\{\boldsymbol{v}_i\}$. They fail to make use of the past queries' responses to construct the new query and recover the full gradient more efficiently. As stated in the introduction, we solve the smaller problem of *gradient sign estimation* with adaptive queries based on the observation that simply leveraging (noisy) sign bits of the gradient yields successful attacks–see Figure 1.

**Definition 1.** *(Gradient Sign Estimation Problem) For an input/label pair $(\boldsymbol{x}, y)$ and a loss function $L$, let $\boldsymbol{g}^* = \nabla_{\boldsymbol{x}} L(\boldsymbol{x}, y)$ be the gradient of $L$ at $(\boldsymbol{x}, y)$ and $\boldsymbol{q}^* = \mathrm{sign}(\boldsymbol{g}^*) \in \mathcal{H}$ be the sign bit vector of $\boldsymbol{g}^*$.[2] Then the goal of the gradient sign estimation problem is to find a binary vector $\boldsymbol{q} \in \mathcal{H}$ maximizing the directional derivative[3]*

$$\max_{\boldsymbol{q} \in \mathcal{H}} D_{\boldsymbol{q}} L(\boldsymbol{x}, y) \ , \tag{2}$$

*from a limited number of (possibly adaptive) function value queries $L(\boldsymbol{x}', y)$.*

## 3    A METHOD FOR ESTIMATING SIGN OF THE GRADIENT FROM ADAPTIVE QUERIES

Our goal is to estimate the gradient sign bits of the loss function $L$ of the model under attack at an input/label pair $(\boldsymbol{x}, y)$ from a limited number of loss value adaptive queries $L(\boldsymbol{x}', y)$. To this end, we examine the basic concept of directional derivatives that has been employed in recent black-box

---

[2]Without loss of generality, we encode the sign bit vector in $\mathcal{H} \equiv \{-1, +1\}^n$ rather than $\{0, 1\}^n$. This is a common representation in sign-related literature. Note that the standard sign function has the range of $\{-1, 0, +1\}$. Here, we use the non-standard definition (Zhao, 2018) whose range is $\{-1, +1\}$. This follows from the observation that DNNs' gradients with respect to their inputs are not sparse (Ilyas et al., 2019, Appendix B.1).

[3]The maximization follows from $D_{\boldsymbol{q}} L(\boldsymbol{x}, y) = \boldsymbol{q}^T \boldsymbol{g}^*$, which is maximized when $\boldsymbol{q} = \boldsymbol{q}^* = \mathrm{sign}(\boldsymbol{g}^*)$.

adversarial attacks. Based on the definition of the directional derivative (Eq. 1), the following can be stated.

**Property 1** (Separability of $D_{\boldsymbol{q}}L(\boldsymbol{x}, y)$)**.** *The directional derivative $D_{\boldsymbol{q}}L(\boldsymbol{x}, y)$ of the loss function $L$ at an input/label pair $(\boldsymbol{x}, y)$ in the direction of a binary code $\boldsymbol{q}$ is separable. That is,*

$$\max_{\boldsymbol{q} \in \mathcal{H}} D_{\boldsymbol{q}}L(\boldsymbol{x}, y) = \max_{\boldsymbol{q} \in \mathcal{H}} \boldsymbol{q}^T \boldsymbol{g}^* = \sum_{i=1}^{n} \max_{q_i \in \{-1, +1\}} q_i g_i^* \; . \tag{3}$$

This reformulates the gradient sign estimation problem from single $n$-dimensional to $n$ 1-dimensional binary black-box optimization problems, reducing the search space of sign bits from $2^n$ to $2n$. Subsequently, one could recover the gradient sign bits with $n + 2$ queries as follows: i. Start with an arbitrary sign vector $\boldsymbol{q}$ and compute the directional derivative $D_{\boldsymbol{q}}L(\boldsymbol{x}, y)$. Using Eq. 1, this requires two queries: $L(\boldsymbol{x} + \delta\boldsymbol{q}, y)$ and $L(\boldsymbol{x}, y)$. ii. For the remaining $n$ queries, flip $\boldsymbol{q}$'s bits (coordinates) one by one and compute the corresponding directional derivative–one query each $L(\boldsymbol{x} + \delta\boldsymbol{q}, y)$. iii. Retain bit flips that maximize the directional derivative $D_{\boldsymbol{q}}L(\boldsymbol{x}, y)$ and revert those otherwise. This, however, still suffers from the $O(n)$ complexity of *full* gradient estimation methods. Further, each query recovers at most one sign bit and the natural question to ask is: can we recover more sign bits per query?

Consider the case where all the gradient coordinates have the same magnitude, i.e., $|\{|g_i^*|\}_{1 \leq i \leq n}| = 1$, and let the initial guess $\boldsymbol{q}_1$ have $r$ correct bits and $n - r$ wrong ones. Instead of flipping its bits sequentially, we can flip them all at once to get $\boldsymbol{q}_2 = -\boldsymbol{q}_1$. If $D_{\boldsymbol{q}_2}L(\boldsymbol{x}, y) \geq D_{\boldsymbol{q}_1}L(\boldsymbol{x}, y)$, then we retain $\boldsymbol{q}_2$ as our best guess with $n - r$ correct bits, otherwise $\boldsymbol{q}_1$ remains. In either cases, with three queries, we will recover $\max(r, n - r)$ sign bits. One can think of this *flip/revert* procedure as one of *majority voting*

---

**Algorithm 1** `SignHunter`
$g : \mathcal{H} \to \mathbb{R}$ : the black-box function to be maximized over the binary hypercube $\mathcal{H} \equiv \{-1, +1\}^n$

> **def** init($g$) :
>     $i \leftarrow 0, h \leftarrow 0$
>     $g \leftarrow g$
>     $\boldsymbol{s} \sim \mathcal{U}(\mathcal{H})$        // e.g., $[+1, \ldots, +1]$
>     done $\leftarrow false$
>     $g_{best} \leftarrow -\infty$
>
> **def** is_done() :
>     return done
>
> **def** step() :
>     c_len $\leftarrow \lceil n/2^h \rceil$
>     $\boldsymbol{s}$[i*c_len:(i+1)*c_len] *= -1
>     if $g(\boldsymbol{s}) \geq g_{best}$:
>         $g_{best} \leftarrow g(\boldsymbol{s})$
>     else:
>         $\boldsymbol{s}$[i*c_len:(i+1)*c_len] *= -1
>     increment $i$
>     if $i == 2^h$:
>         $i \leftarrow 0$, increment $h$
>         if $h == \lceil \log_2(n) \rceil + 1$:
>             done $\leftarrow true$
>
> **def** get_current_sign_estimate() :
>     return $\boldsymbol{s}$

---

by the guess's coordinates on whether they agree with their gradient sign's counterparts. To see this, let $|g_i^*| = 1$ for all $i$, then the condition $D_{\boldsymbol{q}_2}L(\boldsymbol{x}, y) \geq D_{\boldsymbol{q}_1}L(\boldsymbol{x}, y)$ can be written as $n - r - r \geq r - n + r \implies n \geq 2r$. If the *agree* votes $r$ are less than half of the total votes $n$, then $\boldsymbol{q}_2$ is retained. Besides flipping *all* the coordinates, one can employ the same procedure iteratively on a subset (chunk) of the coordinates $[q_j, \ldots, q_{j+n_i}]$ of the guess vector $\boldsymbol{q}$, recovering $\max(r_i, n_i - r_i)$ sign bits, where $n_i$ and $r_i$ is the length of the $i$th chunk and the number of its correct signs, respectively.

While the magnitudes of gradient coordinates may not have the same value as assumed in the previous example; through empirical evaluation (see Appendix F), we found them to be concentrated. Consequently and with high probability, their votes on retaining or reverting chunks of sign flips are weighted (by their corresponding gradient magnitude) similarly. That said, if we are at a chunk where the distribution of the gradient coordinate magnitudes is uniform, then the flip/revert procedure could favor recovering *few* sign coordinates with large magnitude counterparts over *many* sign coordinates with small magnitude counterparts. From our experiments on the noisy `FGSM`, this still suffices to generate adversarial examples: an attack with $30\%$ correct sign bits (that correspond to the top gradient coordinates magnitudes) is more effective than an attack with $50\%$ correct *arbitrary* sign bits as shown in Figure 1. Put differently, we would like to recover as many sign bits as possible with as few queries as possible. However, if we can only recover few, they should be those that correspond to coordinates with large gradient magnitude. This notion is in line with the flip/revert procedure.

We employ the above observation in a divide-and-conquer search which we refer to as `SignHunter`. As outlined in Algorithm 1, the technique starts with an initial guess of the sign vector $\boldsymbol{q}_1$ ($\boldsymbol{s}$ in Algorithm 1). It then proceeds to flip the sign of all the coordinates to get a new sign vector $\boldsymbol{q}_2$,

and revert the flips if the loss oracle returned a value $L(\boldsymbol{x} + \delta\boldsymbol{q}_2, y)$ (or equivalently the directional derivative ) less than the best obtained so far $L(\boldsymbol{x} + \delta\boldsymbol{q}_1, y)$. `SignHunter` applies the same rule to the first half of the coordinates, the second half, the first quadrant, the second quadrant, and so on. For a search space of dimension $n$, `SignHunter` needs $2^{\lceil \log(n)+1 \rceil} - 1$ sign flips to complete its search. If the query budget is not exhausted by then, one can update $\boldsymbol{x}$ with the recovered signs and restart the procedure at the updated point with a new starting code $\boldsymbol{s}$. If we start with a sign vector whose Hamming distance to the optimal sign vector $\boldsymbol{q}^*$ is $n/2$: agreeing with $\boldsymbol{q}^*$ in the first half of coordinates. In this case, `SignHunter` needs just *four* queries to recover the entire sign vector independent of $n$, whereas the sequential bit flipping still require $n + 2$ queries. In the next theorem, we show that `SignHunter` is guaranteed to perform at least as well as `FGSM` with $O(n)$ oracle queries. Up to our knowledge, no such guarantees exist for any black-box attack from the literature.

**Theorem 1.** *(Optimality of* `SignHunter`*) Given* $2^{\lceil \log(n)+1 \rceil}$ *queries and that the directional derivative is well approximated by the finite-difference (Eq. 1),* `SignHunter` *is at least as effective as* `FGSM` *(Goodfellow et al., 2015) in crafting adversarial examples.*

The proof can be found in Appendix B. Theorem 1 provides an upper bound on the number of queries required for `SignHunter` to recover the gradient sign bits, and perform as well as `FGSM`. In practice (as will be shown in our experiments), `SignHunter` crafts adversarial examples with a small fraction of this upper bound. The rationale here is that we do not need to recover the sign bits exactly; we rather need a fast convergence to an *adversarially helpful* sign vector $\boldsymbol{s}$. In our setup, we use the best sign estimation obtained $\boldsymbol{s}$ so far in a similar fashion to `FGSM`, whereas full-gradient estimation approaches often employ an iterative scheme of $T$ steps within the perturbation ball $B_p(\boldsymbol{x}, \epsilon)$, calling the *gradient estimation routine* in every step leading to a search complexity of $nT$. Instead, our *gradient sign estimation routine* runs at the top level of our adversarial example generation procedure. Further, `SignHunter` is amenable to parallel hardware architecture and has a smaller memory footprint (just sign bits) and thus can carry out attacks in batches more efficiently. Crafting black-box adversarial attacks with `SignHunter` is outlined in Algorithm 2.

## 4 EXPERIMENTS

We evaluate `SignHunter` and compare it with established algorithms from the literature: `ZO-SignSGD` Liu et al. (2019), `NES` Ilyas et al. (2018), and `Bandits`$_{TD}$ Ilyas et al. (2019) in terms of effectiveness in crafting (without loss of generality) untargeted black-box adversarial examples. To highlight `SignHunter`'s adaptive query construction, we introduce a variant of Algorithm 2, named `Rand`. At every iteration, `Rand`'s sign vector is sampled uniformly from $\mathcal{H}$.[4] Both $\ell_\infty$ and $\ell_2$ threat models are considered on the MNIST, CIFAR10, and IMAGENET datasets. Code and data for the experiments can be found at https://bit.ly/3acIHoQ.

---

**Algorithm 2** Black-Box Adversarial Example Generation with `SignHunter`

$\boldsymbol{x}_{init}$: input to be perturbed, $y_{init}$ : $\boldsymbol{x}_{init}$'s true label,
$B_p(., \epsilon)$ : $\ell_p$ perturbation ball of radius $\epsilon$
$L$ : loss function of the model under attack

---

1:   $\delta \leftarrow \epsilon$ // set finite-difference probe to perturbation bound
2:   $\boldsymbol{x}_o \leftarrow \boldsymbol{x}_{init}$
3:   Define the function $g$ as

$$g(\boldsymbol{q}) = \frac{L(\Pi_{B_p(\boldsymbol{x}_{init},\epsilon)}(\boldsymbol{x}_o + \delta\boldsymbol{q}), y_{init}) - L(\boldsymbol{x}_o, y_{init})}{\delta}$$

4:   `SignHunter`.init($g$)
5:   // $C(\cdot)$ returns top class
6:   **while** $C(\boldsymbol{x}) = y_{init}$ **do**
7:      `SignHunter`.step()
8:      $\boldsymbol{s} \leftarrow$ `SignHunter`.get_current_sign_estimate()
9:      $\boldsymbol{x} \leftarrow \Pi_{B_p(\boldsymbol{x}_{init},\epsilon)}(\boldsymbol{x}_o + \delta\boldsymbol{s})$
10:      **if** `SignHunter`.is_done() **then**
11:         $\boldsymbol{x}_o \leftarrow \boldsymbol{x}$
12:         Define the function $g$ as in Line 3 (with $\boldsymbol{x}_o$ update)
13:         `SignHunter`.init($g$)
14: **return** $\boldsymbol{x}$

---

***Experiments Setup.*** Our experiment setup is similar to (Ilyas et al., 2019). Each attacker is given a budget of $10,000$ oracle queries per attack attempt and is evaluated on 1000 images from the test sets of MNIST, CIFAR10, and the validation set of IMAGENET. We did not find a standard practice for setting the perturbation bound $\epsilon$. For the $\ell_\infty$ threat model, we use (Madry et al., 2017)'s bound for MNIST and (Ilyas et al., 2019)'s bounds for both CIFAR10 and IMAGENET. For the $\ell_2$ threat model, (Ilyas et al., 2019)'s bound is used for IMAGENET. MNIST's bound is set based on the sufficient distortions observed in (Liu et al.,

---

[4]That is, replace Line 8 in Algorithm 2 by $\boldsymbol{s} \sim \mathcal{U}(\mathcal{H})$.

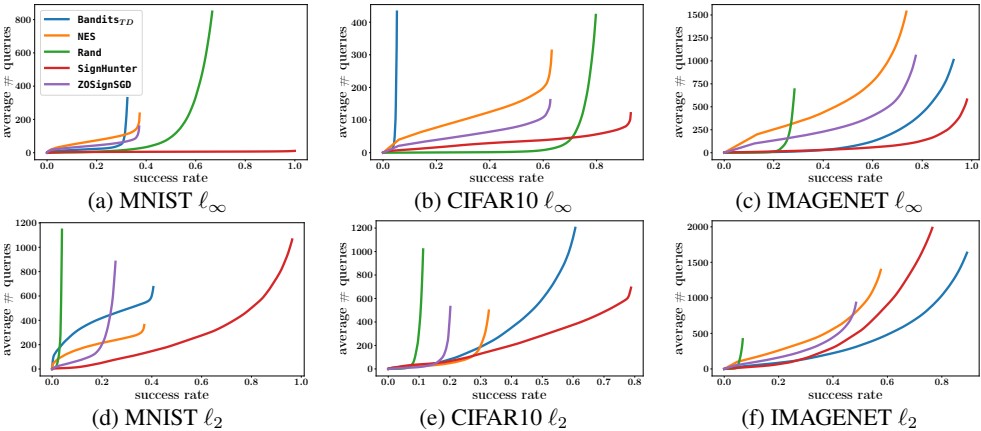

Figure 2: Performance of black-box attacks in the $\ell_\infty$ and $\ell_2$ perturbation constraint. The plots show the average number of queries used per successful image for each attack when reaching a specified success rate.

2019), which are smaller than the one used in (Madry et al., 2017). We use the observed bound in (Cohen et al., 2019) for CIFAR10. We show results based on standard models–i.e., models that are not adversarially hardened. For MNIST and CIFAR10, the naturally trained models from (Madry et al., 2017)'s MNIST and CIFAR10 challenges are used. For IMAGENET, TensorFlow's Inception (v3) model is used. The loss oracle returns the cross-entropy loss of the respective model. See Appendix C for other general experimental setup details.

***Hyperparameters Setup.*** While SignHunter does not have any hyperparameters, to fairly compare it with the other algorithms, we tuned their hyperparameters starting with the default values reported by the corresponding authors. The finite difference probe $\delta$ for SignHunter is set to the perturbation bound $\epsilon$ as it is used for both computing the finite difference and crafting the adversarial examples— see Line 1 in Algorithm 2. This tuning-free aspect of SignHunter offers a robustness advantage over algorithms which require expert hypertuning. Details on the hyperparameter setup are available in Appendix C.

Table 1: Summary of attacks effectiveness on CIFAR10 under $\ell_\infty$ and $\ell_2$ perturbation constraints, and with a query limit of 10, 000 queries. The *Failure Rate* $\in [0, 1]$ column lists the fraction of failed attacks over 1000 images. The *Avg. # Queries* column reports the average number of queries made to the loss oracle only over successful attacks.

| | Failure Rate | | Avg. # Queries | |
| | $\ell_\infty$ | $\ell_2$ | $\ell_\infty$ | $\ell_2$ |
| **Attack** | | | | |
| --- | --- | --- | --- | --- |
| Bandits$_{TD}$ | 0.95 | 0.39 | 432.24 | 1201.85 |
| NES | 0.37 | 0.67 | 312.57 | **496.99** |
| Rand | 0.20 | 0.89 | 422.16 | 1018.17 |
| SignHunter | **0.07** | **0.21** | **121.00** | 692.39 |
| ZOSignSGD | 0.37 | 0.80 | 161.28 | 528.35 |

***Results.*** Figure 2 shows the trade-off between the success (evasion) rate and the mean number of queries (of the successful attacks, per convention) needed to generate an adversarial example for the MNIST, CIFAR10, and IMAGENET classifiers under the $\ell_\infty$ and $\ell_2$ perturbation constraints. These plots indicate the average number of queries required for a desired success rate. Table 1 represents a tabulated summary of plots (b) and (e) of Figure 2.[5] We observe the following: For any given success rate, SignHunter dominates the previous state of the art approaches in all settings except the IMAGENET $\ell_2$ setup, where Bandits$_{TD}$ shows a better query efficiency when the desired success rate is roughly greater than 0.35. This is all the more remarkable because Bandits$_{TD}$

---

[5]More tabulated summaries of these plots can be found in Appendix D–Tables 7–9. We also plot the classifier's loss and the gradient estimation quality (in terms of Hamming distance and cosine similarity) averaged over all the images as a function of the number of queries in Figures 4–6 of Appendix D.

exploits *tiles*, a data-dependent prior, searching over $50 \times 50 \times 3$ dimensions for IMAGENET, while `SignHunter` searches over the explicit data $299 \times 299 \times 3$ dimensions: $36\times$ more dimensions.

$\ell_\infty$ *vs.* $\ell_2$ *Perturbation Threat.* In view of `Bandits`$_{TD}$'s advantage, `SignHunter` is remarkably efficient in the $\ell_\infty$ setup, achieving a **100%** evasion using—on average—just **12** queries per image against the MNIST classifier! In the $\ell_2$ setup, `SignHunter`'s performance degrades—yet it still outperforms the other algorithms. This is expected, since `SignHunter` perturbs all the coordinates with the same magnitude and the $\ell_2$ perturbation bound $\epsilon_2$ for all the datasets in our experiments is set such that $\epsilon_2/\sqrt{n}$ is significantly less than the $\ell_\infty$ perturbation bound $\epsilon_\infty$. Take the case of MNIST ($n = 28 \times 28$), where $\epsilon_\infty = 0.3$ and $\epsilon_2 = 3$. For `SignHunter`, the $\ell_2$ setup is equivalent to an $\ell_\infty$ perturbation bound of $3/28 \approx 0.1$. The employed $\ell_2$ perturbation bounds give the state of the art—continuous optimization based—approaches more perturbation options. For instance, it is possible for `NES` to perturb just one pixel in an MNIST image by a magnitude of 3; two pixels by a magnitude of $3/\sqrt{2} \approx 2.1$ each; ten pixels by a magnitude of $3/\sqrt{10} \approx 0.9$ each, etc. On the other hand, the binary optimization view of `SignHunter` limits it to always perturb all $28 \times 28$ pixels by a magnitude of $3/28 \approx 0.1$. Despite its fewer degrees of freedom, `SignHunter` maintains its effectiveness in the $\ell_2$ setup. The plots can also be interpreted as a sensitivity assessment of `SignHunter` as $\epsilon$ gets smaller going from $\ell_\infty$ to the $\ell_2$ perturbation threat.

*SignHunter vs FGSM.* The performance of `SignHunter` is in line with Theorem 1 when compared with the performance of `FGSM` (the noisy `FGSM` at $k = 100\%$ in Figures 1 and 2 of Appendix A) in both $\ell_\infty$ and $\ell_2$ setups across all datasets. For instance, `FGSM` has a failure rate of $0.32$ for CIFAR10 $\ell_2$ (Appendix A, Figure 2 (b)), while `SignHunter` achieves a failure rate of $0.21$ with $692.39 < 2n = 2 \times 3 \times 32 \times 32 = 6144$ queries (Appendix D, Table 8). Note that for IMAGENET, `SignHunter` outperforms `FGSM` with a query budget of $10,000$ queries, a fraction of the theoretical number of queries required $2n = 536,406$ to perform at least as well. Incorporating `SignHunter` in an iterative framework of perturbing the data point $\boldsymbol{x}$ till the query budget is exhausted (Lines 10 to 14 in Algorithm 2) supports the observation in white-box settings that iterative `FGSM`—or Projected Gradient Descent (`PGD`)—is stronger than `FGSM` (Madry et al., 2017; Al-Dujaili et al., 2018). This is evident by the upticks in `SignHunter`'s performance on the MNIST $\ell_2$ case (Appendix D, Figure 4), which happens after every iteration (after every other $2 \times 28 \times 28$ queries).

*Gradient Estimation.* Plots of the Hamming similarity capture the number of recovered sign bits, while plots of the average cosine similarity capture the value of Eq. 2. Both `SignHunter` and `Bandits`$_{TD}$ consistently optimize both metrics. In general, `SignHunter` (`Bandits`$_{TD}$) converges faster especially on the Hamming(cosine) metric as it is estimating the signs(signs and magnitudes) compared to `Bandits`$_{TD}$'s full gradient (`SignHunter`'s gradient sign) estimation. This is most obvious in the IMAGENET $\ell_2$ setup (Appendix D, Figure 6). Note that once an attack is successful, the estimated gradient sign at that point is used for the rest of the plot. This explains why, in the $\ell_\infty$ settings, `SignHunter`'s plot does not improve compared to its $\ell_2$ counterpart, as most of the attacks are successful in the very first few queries made to the loss oracle and no further refined estimation is required. Another possible reason is that the gradient direction can be very local and does not capture the global loss landscape compared to `SignHunter`'s estimation. More on this is discussed in Section 6.

*SignHunter vs. Rand.* Given these results, one could argue that `SignHunter` is effective, because it maximally perturbs datapoints to the vertices of their perturbation balls.[6] However, `Rand`'s poor performance does not support this argument and highlights the effectiveness of `SignHunter`'s adaptive query construction. Except for MNIST and CIFAR10 $\ell_\infty$ settings, `Rand` performs worse than the full-gradient estimation approaches, although it perturbs datapoints similar to `SignHunter`. Overall, `SignHunter` is $3.8\times$ less failure-prone than the state-of-the-art approaches combined, and spends over all the images (successful and unsuccessful attacks) $2.5\times$ less queries.[7]

## 5 SignHunter vs. Defenses

To complement Section 4, we evaluate `SignHunter` against *adversarial training*, a way to improve the robustness of DNNs (Madry et al., 2017). Specifically, we attacked the *secret* models used

---

[6] We define perturbation vertices as extreme points of the ball $B_p(\boldsymbol{x}, \epsilon)$. That is, $\boldsymbol{x} \pm \epsilon_\infty$, where $\epsilon_\infty = \epsilon$ when $p = \infty$ and $\epsilon_\infty = \epsilon/\sqrt{n}$ when $p = 2$.

[7] The number of queries spent is computed based on Tables 7–9 of Appendix D as `(1 - fail_rate) * avg_#_queries + fail_rate * 10,000`.

in public challenges for MNIST and CIFAR10. For IMAGENET, we used *ensemble adversarial training*, a method that argues security against black-box attacks based on transferability Tramèr et al. (2017a). Appendix E reports the same metrics used in Section 4 as well as a tabulated summary for the results discussed below.

***Public MNIST Black-Box Attack Challenge.*** In line with the challenge setup, $10,000$ test images were used with an $\ell_\infty$ perturbation bound of $\epsilon = 0.3$. Although the secret model is released, we treated it as a black box similar to our experiments in Section 4. No maximum query budget was specified, so we set it to $5,000$ queries. This is equal to the number of iterations given to a `PGD` attack in the white-box setup of the challenge: 100-steps with 50 random restarts. `SignHunter`'s attacks resulted in the lowest model accuracy of $\mathbf{91.47}\%$, outperforming all the submitted attacks to the challenge, with an average number of queries of $\mathbf{233}$ per successful attack. Note that the attacks submitted to the challenge are based on transferability and do not query the model of interest. On the other hand, the most powerful *white-box* attack by Wang et al. (2018)—as of May 15, 2019—resulted in a model accuracy of $88.42\%$. Further, a `PGD` attack with $5,000$ back-propagations achieves $89.62\%$ in contrast to `SignHunter`'s $91.47\%$ with just $5,000$ forward-propagations.

***Public CIFAR10 Black-Box Attack Challenge.*** This challenge setup is similar to the above, but with an $\ell_\infty$ perturbation bound of $\epsilon = 8$. `SignHunter`'s attacks resulted in the lowest model accuracy of $\mathbf{47.16}\%$, outperforming all the submitted attacks to the challenge, with an average number of queries of $\mathbf{569}$ per successful attack. Similar to the MNIST challenge, all the submitted attacks are based on transferability. On the other hand, the most powerful *white-box* attack by Zheng et al. (2018)—as of May 15, 2019—resulted in a model accuracy of $44.71\%$. Further, a `PGD` attack with 200 back-propagations achieves $45.21\%$ in contrast to `SignHunter`'s $47.16\%$ with $5,000$ forward-propagations.

***Ensemble Adversarial Training on IMAGENET.*** In line with Tramèr et al. (2017a), we set $\epsilon = 0.0625$ and report the v3$_{\text{adv-ens4}}$ model's misclassification over 10,000 random images from IMAGENET's validation set. After 20 queries, `SignHunter` achieves a top-1 error of $40.61\%$ greater than the $33.4\%$ rate of a series of black-box attacks (including `PGD` with 20 iterations) transferred from a substitute model. With 1000 queries, `SignHunter` breaks the model's robustness with a top-1 error of $90.75\%$!

## 6 CHARACTERIZING ADVERSARIAL CONES WITH SIGNHUNTER

Estimating the size of *adversarial cones*, the space of adversarial examples in the vicinity of a point, for a model has been a topic of interest by the machine learning community Tramèr et al. (2017a); Ma et al. (2018); Lu et al. (2018). The Gradient-Aligned Adversarial Subspace (`GAAS`) method Tramèr et al. (2017b) provides an approximation of the adversarial cone dimensionality by finding a set of orthogonal perturbations of norm $\epsilon$ that are all adversarial with respect to the model. By linearizing the model's loss function, this is reduced to finding orthogonal vectors that are maximally aligned with its gradient $\boldsymbol{g}^*$—or its gradient sign $\boldsymbol{q}^*$ in the $\ell_\infty$ setup Tramèr et al. (2017a). In Figure 3, we reproduce (Tramèr et al., 2017a, Fig. 2) and show that aligning the orthogonal vectors with `SignHunter`'s estimation (we refer to this approach as `SAAS`) instead of aligning them with the gradient (`GAAS`) results in a better approximation of the adversarial cone for the two IMAGENET models considered earlier, even when the number of queries given to `SignHunter` is just a fraction of the dimensionality $n$. Through its query-efficient finite-difference sign estimation, `SignHunter` is able to quickly capture the *larger-scale* variation of the loss landscape in the point's neighborhood, rather than the infinitesimal point-wise variation that the gradient provides, which can be *very local*. This is important in adversarial settings, where the loss landscape is analyzed in the vicinity of the point Moosavi-Dezfooli et al. (2018); Tramèr et al. (2017a). One interesting observation at $k = 1$ (note here, $\boldsymbol{r}_1 = \boldsymbol{q}^*$) across all $\epsilon$ is that `GAAS` finds adversarial directions for fewer points against the v3$_{\text{adv-ens4}}$ model than the naturally trained model v3, whereas `SAAS` reports similar probability of adversarial directions for both. This contrast suggests that *ensemble adversarial training* Tramèr et al. (2017a) still exhibits the *gradient masking* effect, where the gradient poorly approximates the global loss landscape.

## 7 CONCLUSION

Assuming a black-box threat model, we studied the problem of generating adversarial examples for neural nets and proposed the gradient *sign* estimation problem as the core challenge in crafting

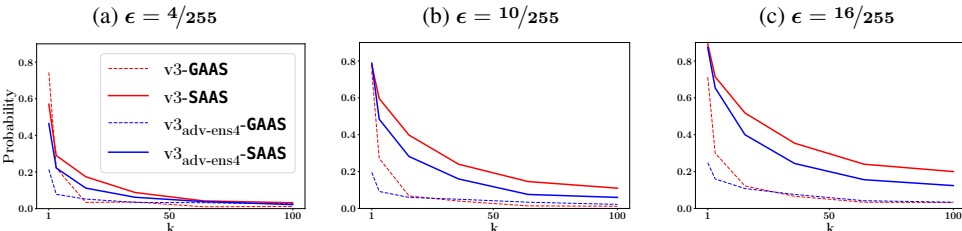

Figure 3: Two estimations of the $\ell_\infty$ adversarial cones for two IMAGENET models: v3 and v3$_{\text{adv-ens4}}$. The first estimation (GAAS: Gradient-Aligned Adversarial Subspace) finds $k$ orthogonal vectors maximally aligned with the gradient sign $q^*$ Tramèr et al. (2017a). The second (SAAS: SignHunter-Aligned Adversarial Subspace) finds $k$ orthogonal vectors that are maximally aligned with SignHunter's $s$ (Algorithm 2, Line 8) after $1,000$ queries. Similar to (Tramèr et al., 2017a, Figure 2), for 500 correctly classified points $x$ and $\epsilon \in \{4, 10, 16\}$, we plot the probability that we find at least $k$ orthogonal vectors $r_i$—computed based on (Tramèr et al., 2017a, Lemma 7)—such that $||r_i||_\infty = \epsilon$ and $x + r_i$ is misclassified. For both models and for the same points $x$, SAAS finds more orthogonal adversarial vectors $r_i$ than GAAS, thereby providing a better characterization of the space of adversarial examples in the vicinity of a point, albeit *without* a white-box access to the models.

these examples. We formulate the problem as a *binary black-box optimization* one: maximizing the directional derivative in the direction of $\{\pm 1\}^n$ vectors, approximated by the finite difference of the queries' loss values. The separability property of the directional derivative helped us devise SignHunter, a query-efficient, tuning-free divide-and-conquer algorithm with a *small* memory footprint that is guaranteed to perform *at least* as well as FGSM after $O(n)$ queries. No similar guarantee is found in the literature. In practice, SignHunter needs a mere fraction of this number of queries to craft adversarial examples. The algorithm is one of its kind to construct *adaptive* queries instead of queries that are based on i.i.d. random vectors. Robust to gradient masking, SignHunter can also be used to estimate the dimensionality of adversarial cones. Moreover, SignHunter achieves the highest evasion rate on two public black-box attack challenges and breaks a model that argues robustness against substitute-model attacks.

### ACKNOWLEDGMENTS

This work was supported by the MIT-IBM Watson AI Lab. We would like to thank Shashank Srikant for his timely help. We are grateful for feedback from Nicholas Carlini and Zico Kolter.

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

## APPENDIX A. NOISY FGSM

This section shows the performance of the noisy FGSM on standard models (described in Section 1 of the main paper) on the MNIST, CIFAR10 and IMAGENET datasets. In Figure 4, we consider the $\ell_\infty$ threat perturbation constraint. Figure 5 reports the performance for the 2 setup. Similar to Ilyas et al. (2019), for each $k$ in the experiment, the top $k$ percent of the signs of the coordinates—chosen either randomly (random-k) or by the corresponding magnitude $|\partial L(\boldsymbol{x}, y)/\partial x_i|$ (top-k)—are set correctly, and the rest are set to $-1$ or $+1$ at random. The misclassification rate shown considers only images that were correctly classified (with no adversarial perturbation). In accordance with the models' accuracy, there were 987, 962, and 792 such images for MNIST, CIFAR10, and IMAGENET out of the sampled 1000 images, respectively. These figures also serve as a validation for Theorem 1 of the main paper when compared to SignHunter's performance shown in Appendix D.

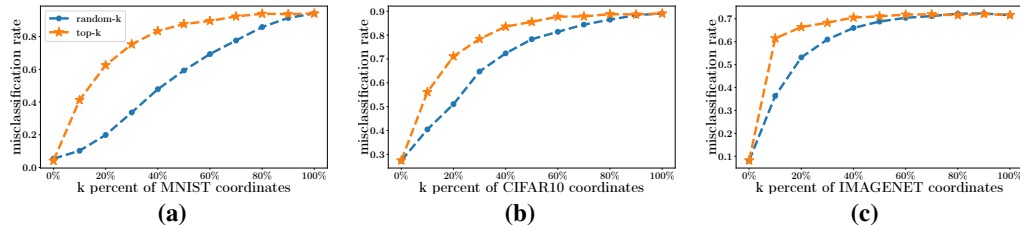

Figure 4: Misclassification rate of three neural nets (for (a) MNIST, (b) CIFAR10, and (c) IMAGENET, respectively) on the *noisy* FGSM's adversarial examples as a function of correctly estimated coordinates of $\text{sign}(\nabla_{\boldsymbol{x}} f(\boldsymbol{x}, y))$ on random 1000 images from the corresponding evaluation dataset, with the maximum allowed $\ell_\infty$ perturbation $\epsilon$ being set to 0.3, 12, and 0.05, respectively. Across all the models, estimating the sign of the top 30% gradient coordinates (in terms of their magnitudes) is enough to achieve a misclassification rate of $\sim 70\%$. Note that Plot (c) is similar to Ilyas et al. (2019)'s Figure 1, but it is produced with TensorFlow rather than PyTorch.

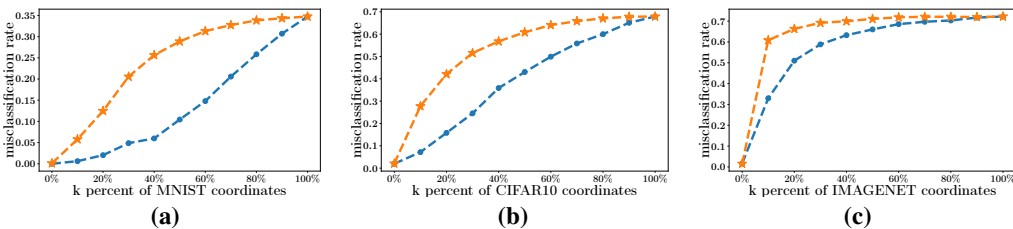

Figure 5: Misclassification rate of three neural nets (for (a) MNIST, (b) CIFAR10, and (c) IMAGENET, respectively) on the *noisy* FGSM's adversarial examples as a function of correctly estimated coordinates of $\text{sign}(\nabla_{\boldsymbol{x}} f(\boldsymbol{x}, y))$ on random 1000 images from the corresponding evaluation dataset, with the maximum allowed $\ell_2$ perturbation $\epsilon$ being set to 3, 127, and 5, respectively. Compared to Figure 4, the performance on MNIST and CIFAR10 drops significantly.

## APPENDIX B. PROOFS FOR THEOREMS IN THE MAIN PAPER

In this section, we present a proof of Theorem 1 of Section 3. Note that the theorem makes the assumption that the finite difference is a good approximation of the directional derivative. This assumption has been the core concept behind most of the black-box adversarial attack algorithms and we state it here for completeness.

**Theorem 1.** *(Optimality of* `SignHunter`*) Given* $2^{\lceil \log(n)+1 \rceil}$ *queries and that the directional derivative is well approximated by the finite-difference (Eq. 1 in the main paper),* `SignHunter` *is at least as effective as* `FGSM` *(Goodfellow et al., 2015) in crafting adversarial examples.*

*Proof.* Based on the separability property of the directional derivative, the $i$th coordinate of the gradient sign vector can be recovered as follows: construct two binary codes $\boldsymbol{u}$ and $\boldsymbol{v}$ such that *only* their $i$th bit is different. Therefore, we have

$$q_i^* = \operatorname{sign}(g_i^*) \quad = \quad \begin{cases} u_i & \text{if } D_{\boldsymbol{u}}L(\boldsymbol{x}, y) > D_{\boldsymbol{v}}L(\boldsymbol{x}, y) \text{ ,} \\ v_i & \text{otherwise .} \end{cases} \tag{4}$$

From the definition of `SignHunter`, this is carried out for all the $n$ coordinates after $2^{\lceil \log(n)+1 \rceil}$ queries. Put it differently, after $2^{\lceil \log(n)+1 \rceil}$ queries, `SignHunter` has flipped every coordinate alone recovering its sign exactly as shown in Eq. 4 above. Therefore, the gradient sign vector is fully recovered, and one can employ the `FGSM` attack to craft an adversarial example. Note that this is under the assumption that our finite difference approximation of the directional derivative (Eq. 1 in the main paper) is good enough (or at least rank-preserving). $\square$

## APPENDIX C. EXPERIMENTS SETUP

This section outlines the experiments setup. To ensure a fair comparison among the considered algorithms, we did our best in tuning their hyperparameters. Initially, the hyperparameters were set to the values reported by the corresponding authors, for which we observed suboptimal performance. We made use of a synthetic concave loss function to efficiently tune the algorithms for each dataset $\times$ perturbation constraint combination. The performance curves on the synthetic loss function using the tuned values of the hyperparameters did show consistency with the reported results from the literature. For instance, we noted that `ZO-SignSGD` converges faster than `NES`, and that $\texttt{Bandits}_{TD}$ outperformed the rest of the algorithms towards the end of query budget. Further, in our adversarial examples generation experiments, we observed failure rate and query efficiency in line with the algorithms' corresponding papers—e.g., compare the performance of $\texttt{Bandits}_{TD}$ and `NES` in Table 9 of Appendix D with (Ilyas et al., 2019, Table 1). That said, we invite the community to provide their best tuned attacks.

Note that `SignHunter` does not have any hyperparameters to tune. The finite difference probe $\delta$ for `SignHunter` is set to the perturbation bound $\epsilon$ as it is used for for both computing the finite difference and crafting the adversarial examples—see Line 1 in Algorithm 2 of the main paper. This tuning-free setup of `SignHunter` offers a robust edge over the state-of-the-art black-box attacks, which often require expert knowledge to carefully tune their parameters.

Table 3 describes the general setup for the experiments. Table 2 lists the sources of the models we attacked in this work, while Tables 4, 5, 6, and 7 outline the algorithms' hyperparameters. Figure 6 shows the performance of the considered algorithms on a synthetic concave loss function after tuning their hyperparameters. All experiments were run on a CUDA-enabled NVIDIA Tesla V100 16GB.

A possible explanation of `SignHunter`'s superb performance is that the synthetic loss function is well-behaved in terms of its gradient given an image. That is, most of gradient coordinates share the same sign, since pixels tend to have the same values and the optimal value for all the pixels is the same $\frac{x_{min}+x_{max}}{2}$. Thus, `SignHunter` will recover the true gradient sign with as few queries as possible (recall the example in Section 3 of the main paper). Moreover, given the structure of the synthetic loss function, the optimal loss value is always at the boundary of the perturbation region; the boundary is where `SignHunter` samples its perturbations.

Table 2: Source of attacked models.

| Model | Source |
|---|---|
| MNIST models | https://github.com/MadryLab/mnist_challenge |
| CIFAR10 models | https://github.com/MadryLab/cifar10_challenge |
| IMAGENET- v3 model | https://bit.ly/2VYDc4X |
| IMAGENET- v3$_{\text{adv-ens4}}$ model | https://bit.ly/2XWTdKx |

Table 3: General setup for all the attacks

| | Value | | | | | |
| | MNIST | | CIFAR10 | | IMAGENET | |
| Parameter | $\ell_\infty$ | $\ell_2$ | $\ell_\infty$ | $\ell_2$ | $\ell_\infty$ | $\ell_2$ |
|---|---|---|---|---|---|---|
| $\epsilon$ (allowed perturbation) | 0.3 | 3 | 12 | 127 | 0.05 | 5 |
| Max allowed queries | | | | 10000 | | |
| Evaluation/Test set size | | | | 1000 | | |
| Data (pixel value) Range | | [0,1] | | [0,255] | | [0,1] |

Table 4: Hyperparameters setup for `NES`

| | Value | | | | | |
| | MNIST | | CIFAR10 | | IMAGENET | |
| Hyperparameter | $\ell_\infty$ | $\ell_2$ | $\ell_\infty$ | $\ell_2$ | $\ell_\infty$ | $\ell_2$ |
|---|---|---|---|---|---|---|
| $\delta$ (finite difference probe) | 0.1 | 0.1 | 2.55 | 2.55 | 0.1 | 0.1 |
| $\eta$ (image $\ell_p$ learning rate) | 0.1 | 1 | 2 | 127 | 0.02 | 2 |
| $q$ (number of finite difference estimations per step) | 10 | 20 | 20 | 4 | 100 | 50 |

Table 5: Hyperparameters setup for `ZO-SignSGD`

| | Value | | | | | |
| | MNIST | | CIFAR10 | | IMAGENET | |
| Hyperparameter | $\ell_\infty$ | $\ell_2$ | $\ell_\infty$ | $\ell_2$ | $\ell_\infty$ | $\ell_2$ |
|---|---|---|---|---|---|---|
| $\delta$ (finite difference probe) | 0.1 | 0.1 | 2.55 | 2.55 | 0.1 | 0.1 |
| $\eta$ (image $\ell_p$ learning rate) | 0.1 | 0.1 | 2 | 2 | 0.02 | 0.004 |
| $q$ (number of finite difference estimations per step) | 10 | 20 | 20 | 4 | 100 | 50 |

Table 6: Hyperparameters setup for `Bandits`$_{TD}$

| | Value | | | | | |
| | MNIST | | CIFAR10 | | IMAGENET | |
| Hyperparameter | $\ell_\infty$ | $\ell_2$ | $\ell_\infty$ | $\ell_2$ | $\ell_\infty$ | $\ell_2$ |
|---|---|---|---|---|---|---|
| $\eta$ (image $\ell_p$ learning rate) | 0.03 | 0.01 | 5 | 12 | 0.01 | 0.1 |
| $\delta$ (finite difference probe) | 0.1 | 0.1 | 2.55 | 2.55 | 0.1 | 0.1 |
| $\tau$ (online convex optimization learning rate) | 0.001 | 0.0001 | 0.0001 | 1e-05 | 0.0001 | 0.1 |
| Tile size (data-dependent prior) | 8 | 10 | 20 | 20 | 50 | 50 |
| $\zeta$ (bandit exploration) | 0.01 | 0.1 | 0.1 | 0.1 | 0.01 | 0.1 |

Table 7: Hyperparameters setup for `SignHunter`

| | Value | | | | | |
| | MNIST | | CIFAR10 | | IMAGENET | |
| Hyperparameter | $\ell_\infty$ | $\ell_2$ | $\ell_\infty$ | $\ell_2$ | $\ell_\infty$ | $\ell_2$ |
|---|---|---|---|---|---|---|
| $\delta$ (finite difference probe) | 0.3 | 3 | 12 | 127 | 0.05 | 5 |

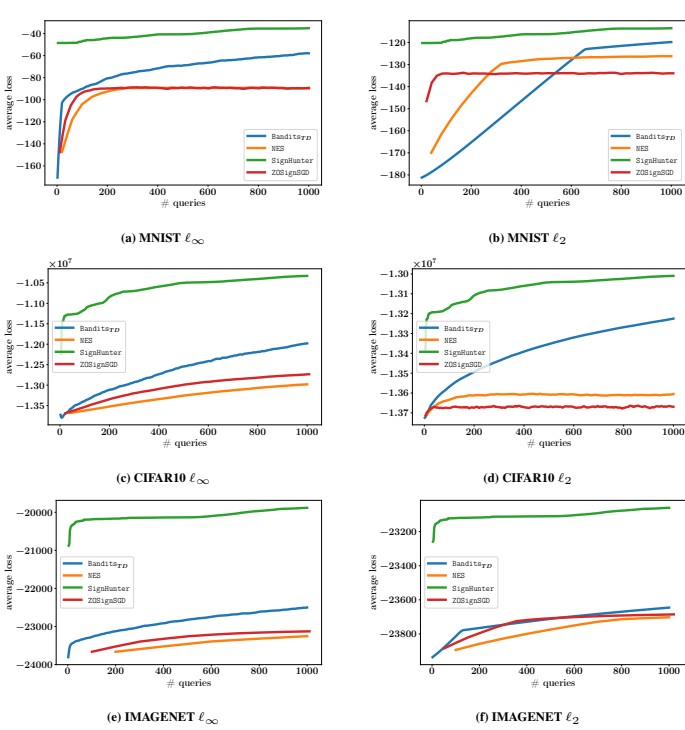

(a) MNIST $\ell_\infty$
(b) MNIST $\ell_2$
(c) CIFAR10 $\ell_\infty$
(d) CIFAR10 $\ell_2$
(e) IMAGENET $\ell_\infty$
(f) IMAGENET $\ell_2$

Figure 6: Tuning testbed for the attacks. A synthetic loss function was used to tune the performance of the attacks over a random sample of 25 images for each dataset and $\ell_p$ perturbation constraint. The plots above show the average performance of the tuned attacks on the synthetic loss function $L(\boldsymbol{x}, y) = -(\boldsymbol{x} - \boldsymbol{x}^*)^T(\boldsymbol{x} - \boldsymbol{x}^*)$, where $\boldsymbol{x}^* = \frac{\boldsymbol{x}_{min} + \boldsymbol{x}_{max}}{2}$ using a query limit of 1000 queries for each image. Note that in all, `Bandits`$_{TD}$ outperforms both `NES` and `ZO-SignSGD`. Also, we observe the same behavior reported by Liu et al. (2019) on the fast convergence of `ZO-SignSGD` compared to `NES`. We did not tune `SignHunter`; it *does not* have any tunable parameters.

## APPENDIX D. RESULTS OF ADVERSARIAL BLACK-BOX EXAMPLES GENERATION

This section shows results of our experiments in crafting adversarial black-box examples on standard models in the form of tables and performance traces, namely Figures 7, 8, and 9; and Tables 8, 9, and 10.

Table 8: Summary of attacks effectiveness on MNIST under $\ell_\infty$ and $\ell_2$ perturbation constraints, and with a query limit of $10,000$ queries. The *Failure Rate* $\in [0, 1]$ column lists the fraction of failed attacks over 1000 images. The *Avg. # Queries* column reports the average number of queries made to the loss oracle only over successful attacks.

| | Failure Rate | | Avg. # Queries | |
| | $\ell_\infty$ | $\ell_2$ | $\ell_\infty$ | $\ell_2$ |
| **Attack** | | | | |
| --- | --- | --- | --- | --- |
| Bandits$_{TD}$ | 0.68 | 0.59 | 328.00 | 673.16 |
| NES | 0.63 | 0.63 | 235.07 | **361.42** |
| Rand | 0.33 | 0.96 | 847.77 | 1144.74 |
| SignHunter | **0.00** | **0.04** | **11.06** | 1064.22 |
| ZOSignSGD | 0.63 | 0.75 | 157.00 | 881.08 |

Table 9: Summary of attacks effectiveness on CIFAR10 under $\ell_\infty$ and $\ell_2$ perturbation constraints, and with a query limit of $10,000$ queries. The *Failure Rate* $\in [0, 1]$ column lists the fraction of failed attacks over 1000 images. The *Avg. # Queries* column reports the average number of queries made to the loss oracle only over successful attacks.

| | Failure Rate | | Avg. # Queries | |
| | $\ell_\infty$ | $\ell_2$ | $\ell_\infty$ | $\ell_2$ |
| **Attack** | | | | |
| --- | --- | --- | --- | --- |
| Bandits$_{TD}$ | 0.95 | 0.39 | 432.24 | 1201.85 |
| NES | 0.37 | 0.67 | 312.57 | **496.99** |
| Rand | 0.20 | 0.89 | 422.16 | 1018.17 |
| SignHunter | **0.07** | **0.21** | **121.00** | 692.39 |
| ZOSignSGD | 0.37 | 0.80 | 161.28 | 528.35 |

Table 10: Summary of attacks effectiveness on IMAGENET under $\ell_\infty$ and $\ell_2$ perturbation constraints, and with a query limit of $10,000$ queries. The *Failure Rate* $\in [0, 1]$ column lists the fraction of failed attacks over 1000 images. The *Avg. # Queries* column reports the average number of queries made to the loss oracle only over successful attacks.

| | Failure Rate | | Avg. # Queries | |
| | $\ell_\infty$ | $\ell_2$ | $\ell_\infty$ | $\ell_2$ |
| **Attack** | | | | |
| --- | --- | --- | --- | --- |
| Bandits$_{TD}$ | 0.07 | **0.11** | 1010.05 | 1635.55 |
| NES | 0.26 | 0.42 | 1536.19 | 1393.86 |
| Rand | 0.72 | 0.93 | 688.77 | **418.02** |
| SignHunter | **0.02** | 0.23 | **578.56** | 1985.55 |
| ZOSignSGD | 0.23 | 0.52 | 1054.98 | 931.15 |

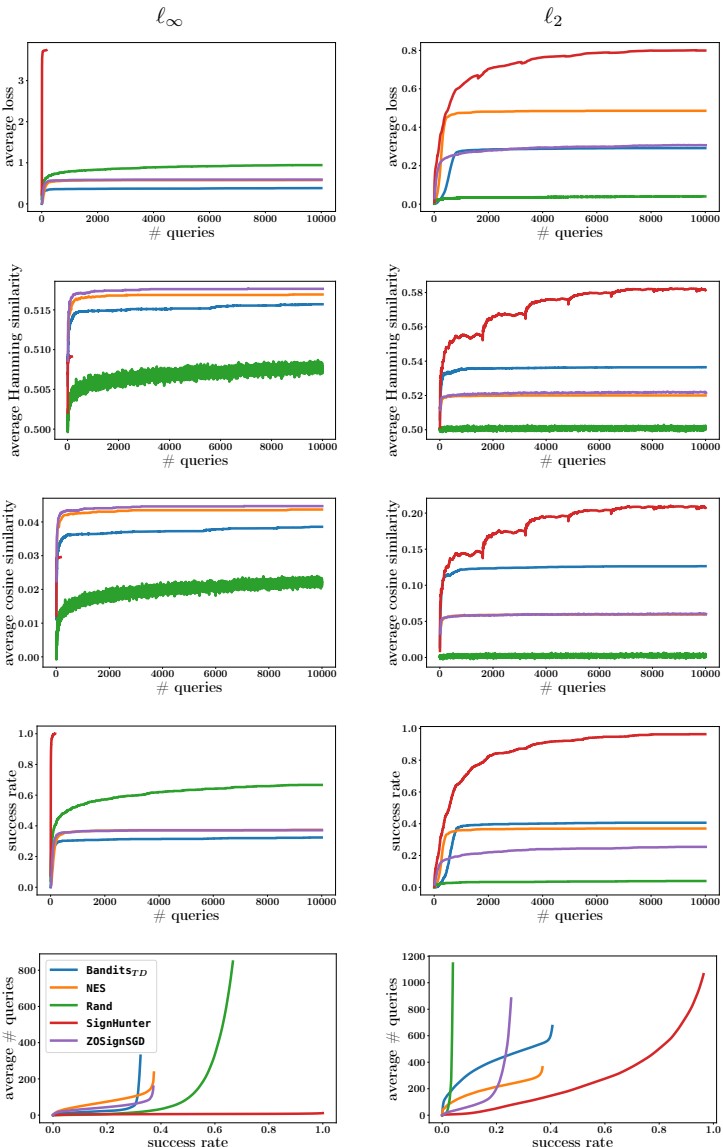

Figure 7: Performance curves of attacks on MNIST for $\ell_\infty$ (first column) and $\ell_2$ (second column) perturbation constraints. Plots of *Avg. Loss* row reports the loss as a function of the number of queries averaged over all images. The *Avg. Hamming Similarity* row shows the Hamming similarity of the sign of the attack's estimated gradient $\hat{g}$ with true gradient's sign $q^*$, computed as $1 - ||\text{sign}(\hat{g}) - q^*||_H / n$ and averaged over all images. Likewise, plots of the *Avg. Cosine Similarity* row show the normalized dot product of $\hat{g}$ and $g^*$ averaged over all images. The *Success Rate* row reports the attacks' cumulative distribution functions for the number of queries required to carry out a successful attack up to the query limit of $10,000$ queries. The *Avg. # Queries* row reports the average number of queries used per successful image for each attack when reaching a specified success rate: the more effective the attack, the closer its curve is to the bottom right of the plot.

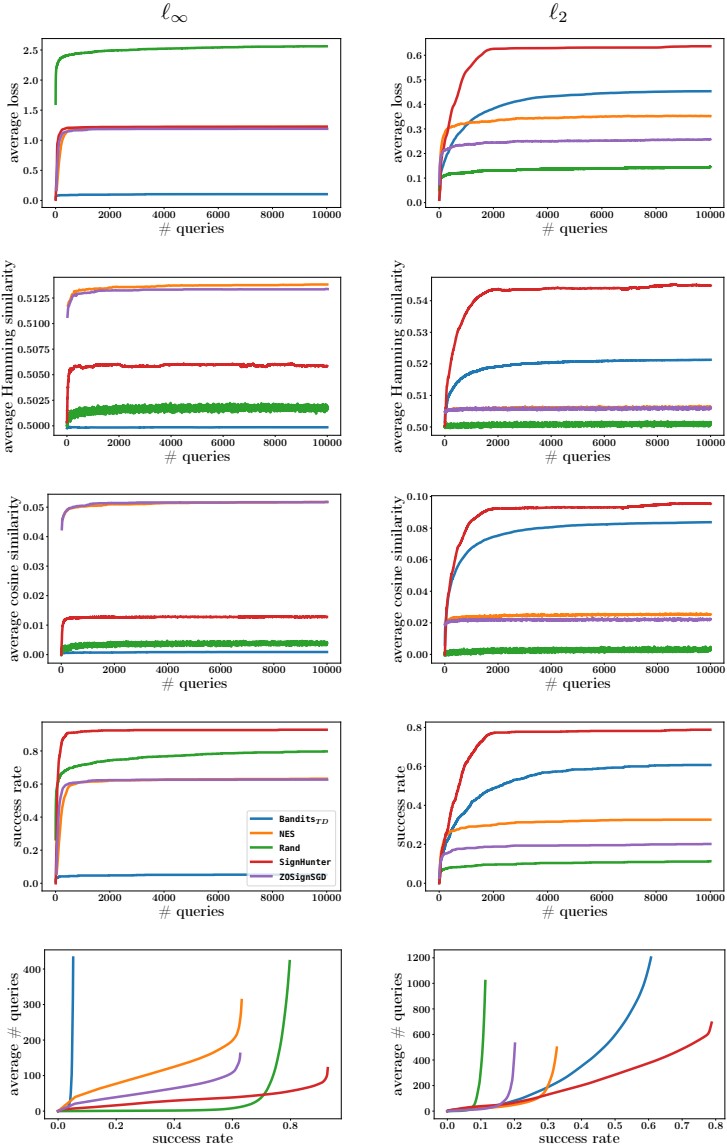

Figure 8: Performance curves of attacks on CIFAR10 for $\ell_\infty$ (first column) and $\ell_2$ (second column) perturbation constraints. Plots of *Avg. Loss* row reports the loss as a function of the number of queries averaged over all images. The *Avg. Hamming Similarity* row shows the Hamming similarity of the sign of the attack's estimated gradient $\hat{g}$ with true gradient's sign $q^*$, computed as $1 - ||\text{sign}(\hat{g}) - q^*||_H/n$ and averaged over all images. Likewise, plots of the *Avg. Cosine Similarity* row show the normalized dot product of $\hat{g}$ and $g^*$ averaged over all images. The *Success Rate* row reports the attacks' cumulative distribution functions for the number of queries required to carry out a successful attack up to the query limit of $10,000$ queries. The *Avg. # Queries* row reports the average number of queries used per successful image for each attack when reaching a specified success rate: the more effective the attack, the closer its curve is to the bottom right of the plot.

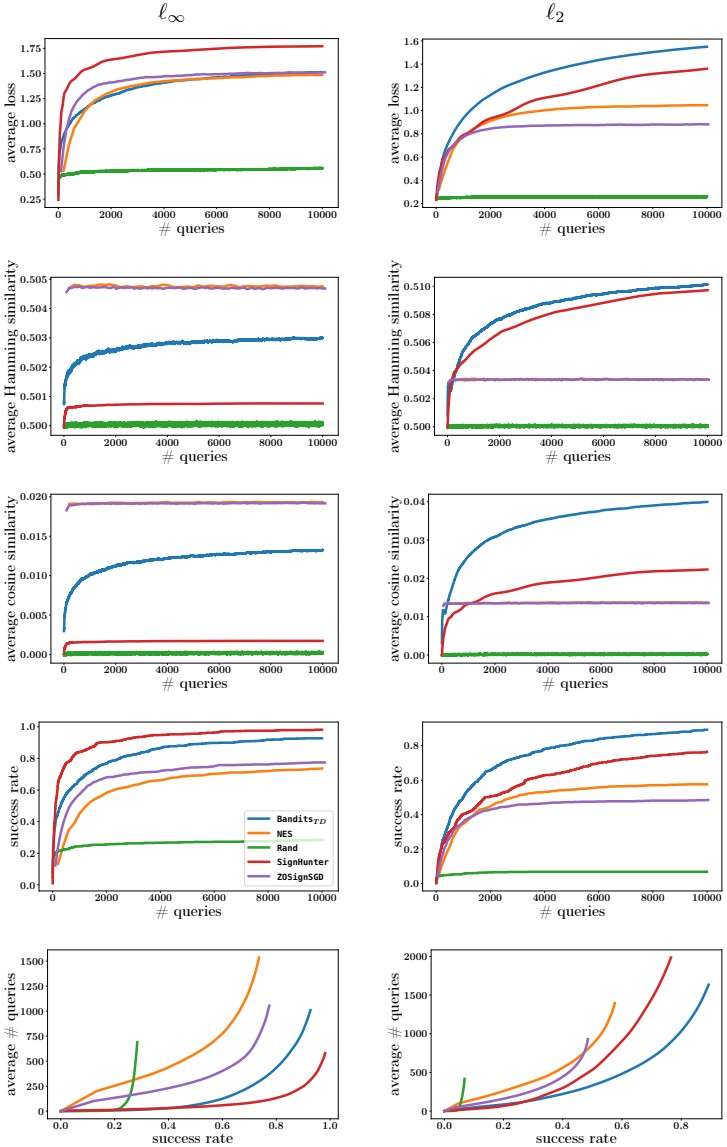

Figure 9: Performance curves of attacks on IMAGENET for $\ell_\infty$ (first column) and $\ell_2$ (second column) perturbation constraints. Plots of *Avg. Loss* row reports the loss as a function of the number of queries averaged over all images. The *Avg. Hamming Similarity* row shows the Hamming similarity of the sign of the attack's estimated gradient $\hat{g}$ with true gradient's sign $q^*$, computed as $1 - ||\text{sign}(\hat{g}) - q^*||_H/n$ and averaged over all images. Likewise, plots of the *Avg. Cosine Similarity* row show the normalized dot product of $\hat{g}$ and $g^*$ averaged over all images. The *Success Rate* row reports the attacks' cumulative distribution functions for the number of queries required to carry out a successful attack up to the query limit of $10,000$ queries. The *Avg. # Queries* row reports the average number of queries used per successful image for each attack when reaching a specified success rate: the more effective the attack, the closer its curve is to the bottom right of the plot.

## APPENDIX E. PUBLIC BLACK-BOX CHALLENGE RESULTS

This section shows results of our experiments in crafting adversarial black-box examples on adversarially trained models in the form of tables and performance traces, namely Tables 11, 12, 13, and Figure 10.

Table 11: Leaderboard of the MNIST black-box challenge. Adapted from the challenge's website—as of May 15, 2019.

| Black-Box Attack | Model Accuracy |
|---|---|
| `SignHunter` (Algorithm 2 in the main paper) | **91.47**% |
| Xiao et al. (2018) | 92.76% |
| `PGD` against 3 independently & adversarially trained copies of the network | 93.54% |
| `FGSM` on the CW loss for model B from (Tramèr et al., 2017a) | 94.36% |
| `FGSM` on the CW loss for the naturally trained public network | 96.08% |
| `PGD` on the cross-entropy loss for the naturally trained public network | 96.81% |
| Attack using Gaussian Filter for selected pixels on the adversarially trained public network | 97.33% |
| `FGSM` on the cross-entropy loss for the adversarially trained public network | 97.66% |
| `PGD` on the cross-entropy loss for the adversarially trained public network | 97.79% |

Table 12: Leaderboard of the CIFAR10 black-box challenge. Adapted from the challenge's website—as of May 15, 2019.

| Black-Box Attack | Model Accuracy |
|---|---|
| `SignHunter` (Algorithm 2 in the main paper) | **47.16**% |
| `PGD` on the cross-entropy loss for the adversarially trained public network | 63.39% |
| `PGD` on the CW loss for the adversarially trained public network | 64.38% |
| `FGSM` on the CW loss for the adversarially trained public network | 67.25% |
| `FGSM` on the CW loss for the naturally trained public network | 85.23% |

Table 13: Top 1 Error Percentage. The numbers between brackets are computed on 10,000 images from the validation set. The rest are from (Tramèr et al., 2017a, Table 4).

| Model | clean | Max. Black-box | `SignHunter` | |
|---|---|---|---|---|
| | | | after 20 queries | after 1000 queries |
| v3$_{\text{adv-ens4}}$ | 24.2 (26.73) | 33.4 | (40.61) | **(90.75)** |

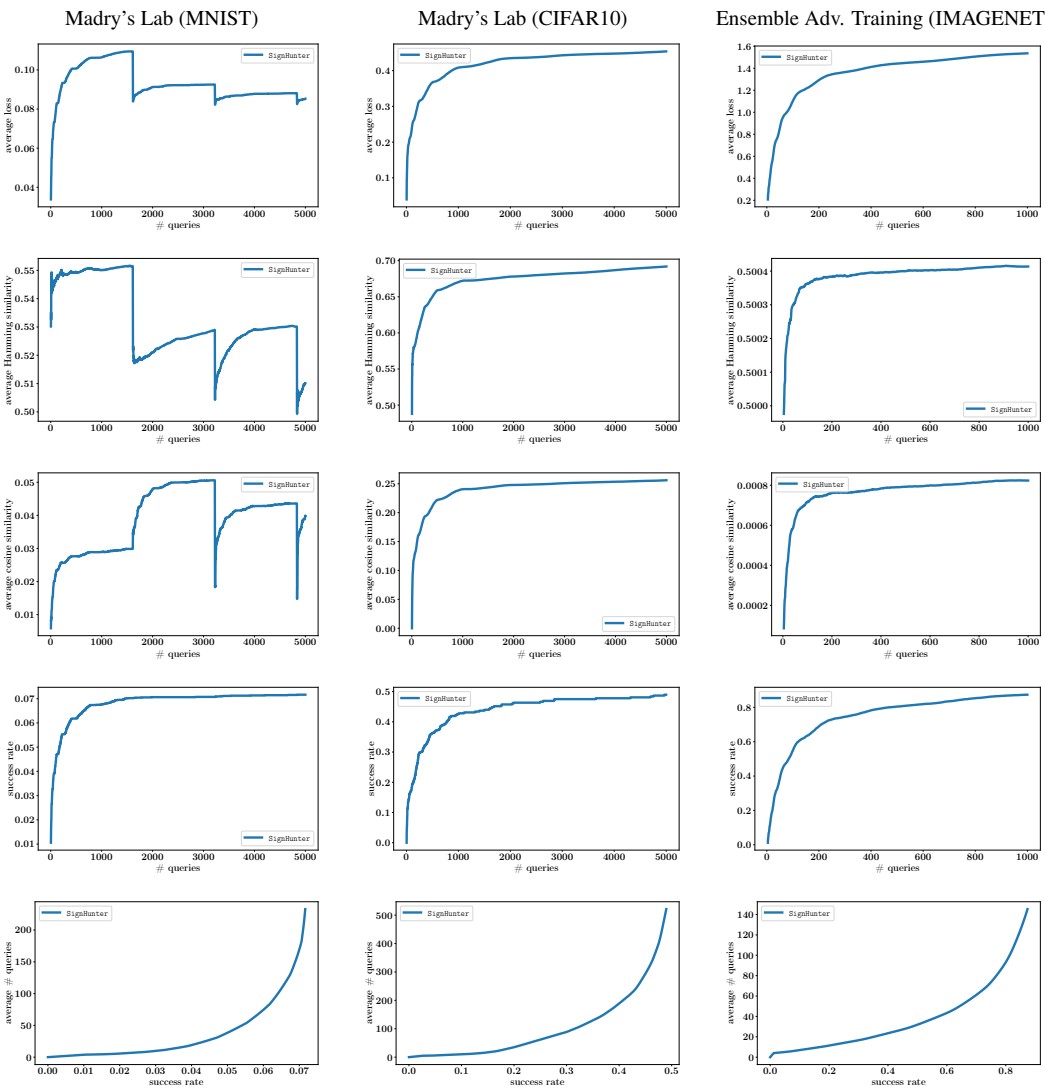

Figure 10: Performance curves of attacks on the public black-box challenges for MNIST (first column), CIFAR10 (second column) and IMAGENET (third column). Plots of *Avg. Loss* row reports the loss as a function of the number of queries averaged over all images. The *Avg. Hamming Similarity* row shows the Hamming similarity of the sign of the attack's estimated gradient $\hat{g}$ with true gradient's sign $q^*$, computed as $1 - ||\text{sign}(\hat{g}) - q^*||_H / n$ and averaged over all images. Likewise, plots of the *Avg. Cosine Similarity* row show the normalized dot product of $\hat{g}$ and $g^*$ averaged over all images. The *Success Rate* row reports the attacks' cumulative distribution functions for the number of queries required to carry out a successful attack up to the query limit of $5,000$ queries for MNIST and CIFAR10 ($1,000$ queries for IMAGENET). The *Avg. # Queries* row reports the average number of queries used per successful image for each attack when reaching a specified success rate: the more effective the attack, the closer its curve is to the bottom right of the plot.

## APPENDIX F. HISTOGRAM OF GRADIENT COORDINATES' MAGNITUDES

This section illustrates our experiment on the distribution of the magnitudes of gradient coordinates as summarized in Figure 11. *How to read the plots:* Consider the first histogram in Plot (a) from below; it corresponds to the $1000^{th}$ image from the sampled MNIST evaluation set, plotting the histogram of the values $\{|\partial L(\boldsymbol{x}, y)/\partial x_i|\}_{1 \le i \le n}$, where the MNIST dataset has dimensionality $n = 784$. These values are in the range $[0, 0.002]$. Overall, the values are fairly concentrated—with exceptions, in Plot (e) for instance, the magnitudes of the $\sim 400^{th}$ image's gradient coordinates are spread from 0 to $\sim 0.055$.

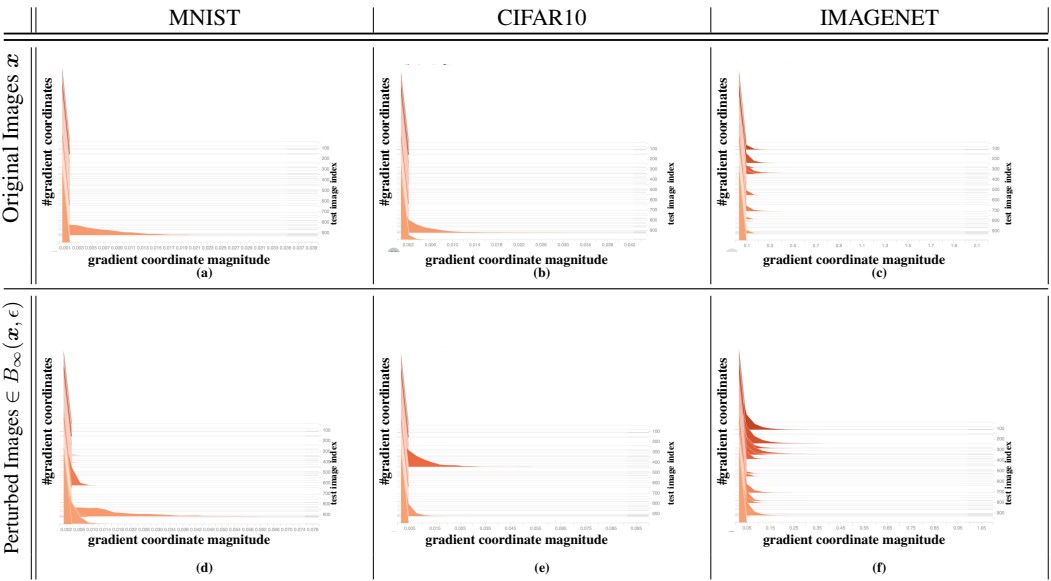

Figure 11: *Magnitudes of gradient coordinates are concentrated:* Plots (a), (b), and (c) show histograms of the magnitudes of gradient coordinates of the loss function $L(\boldsymbol{x}, y)$ with respect to the input point (image) $\boldsymbol{x}$ for MNIST, CIFAR10, and IMAGENET neural net models over 1000 images from the corresponding evaluation set, respectively. Plots (d), (e), (f) show the same but at input points (images) sampled randomly within $B_\infty(\boldsymbol{x}, \epsilon)$: the $\ell_\infty$-ball of radius $\epsilon = 0.3$, 12, and 0.05 around the images in Plots (a), (b), and (c), respectively.

## APPENDIX G. ON SCHEMES FOR SIGN FLIPS

In this section, we show the performance of different sign flip schemes in comparison to SignHunter. Results are summarized in Figure 12. SignHunter's adaptive flips shows a clear advantage over other schemes despite having a worse upper-bound on the query complexity—e.g., Naive can retrieve the signs in $n + 2$ queries, as discussed in Section 3.

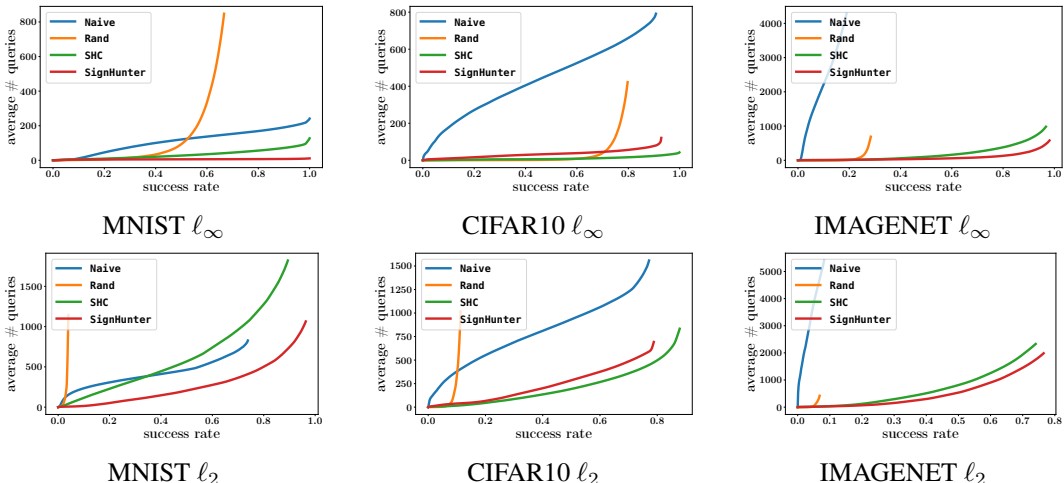

Figure 12: Performance of different sign flips patterns for Algorithm 2, Line 8 in the $\ell_\infty$ and $\ell_2$ perturbation constraints: our proposition (SignHunter), random sign flips (Rand), sequential single sign flips (Naive), stochastic hill climbing (SHC), which is similar to Rand but retain the flip only if it is better in terms of the observed model loss. With higher dimensions, SHC is comparable to SignHunter but does not enjoy a deterministic upper-bound on the query complexity.

## APPENDIX H. SIGNHUNTER AND RECENT RELATED WORK

In this section, we discuss recent work related to our proposition.

***Parsimonious Black-Box Adversarial Attacks (Moon et al., 2019).*** Our experiment on the public CIFAR10 black-box attack challenge corresponds to [1, Table 1]. The authors report a $48\%$ success rate ($52\%$ model accuracy) with an average number of queries of 1261. On the other hand, our proposed algorithm achieves a $52.84\%$ success rate ($47.16\%$ model accuracy) with an average number of queries of 569. Further, (Moon et al., 2019, Table 2) corresponds to our results in Appendix D, Table 9; the paper reports a $98.5\%$ success rate with an average number of queries of 722. Our proposed algorithm achieves a $98\%$ success rate with $578.56$ average number of queries. Based on these numbers, SignHunter demonstrates better performance than (Moon et al., 2019)'s attack.

***Simple Black-Box Attack (SIMBA) (Guo et al., 2019).*** The main distinction is that SIMBA performs a ternary flip over $\{-\delta, 0, +\delta\}$ for one random single coordinate at an iteration with $\delta \leq \epsilon$. On the other hand, SignHunter performs a binary flip $\{-\epsilon, \epsilon\}$ for a group of coordinates at an iteration. Most of Guo et al. (2019)'s experiments were performed for the $\ell_2$ perturbation constraint and against models different from those considered in this paper—except for the IMAGENET v3 model, which the authors find much more difficult to attack. The v3 curves at $10,000$ queries in (Guo et al., 2019, Figure 4) for SIMBA (and its variant SIMBA-DCT) look comparable to SignHunter's of Figure 9. For completeness, we implemented SIMBA and evaluated it against the CIFAR10 model in Section 4. The results are shown in Figure 13. In line with Guo et al. (2019), SIMBA is a strong baseline in the $\ell_2$ setup. However, its performance drops significantly in the $\ell_\infty$ setup.

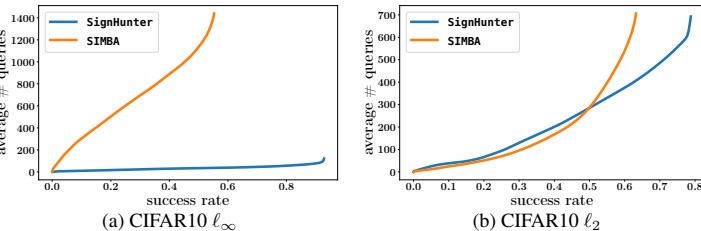

(a) CIFAR10 $\ell_\infty$        (b) CIFAR10 $\ell_2$

Figure 13: Performance of SIMBA and SignHunter in the $\ell_\infty$ and $\ell_2$ perturbation settings of Section 4 on CIFAR10. The plots show the average number of queries used per successful image for each attack when reaching a specified success rate. In line with Guo et al. (2019), we used a step size of $\delta = 50$ for $\ell_2$ (the authors used $\delta = 0.2$ for $[0, 1]$-valued pixels, our setup takes images in $[0, 255]$ so $\delta = 0.2 * 255 \sim 50$). For $\ell_\infty$, we used $\delta = 2$, following NES's setup in Table 4.

***Harmonica (Hazan et al., 2017).*** Both SignHunter and Harmonica seek to optimize a black-box function over the binary hypercube $\{\pm 1\}^n$, albeit with different assumptions on the objective function. Harmonica assumes that the objective function can be approximated by a sparse and low degree polynomial in the Fourier basis. Our assumption with SignHunter is that the objective function is separable (Property 1, Section 3), this lets us optimize the black-box function with $O(n)$ queries given an initial guess instead of searching over the $2^n$ vertices. If this assumption is not met, we can restart SignHunter with another guess with a search complexity of $O(mn)$ where $m$ is the number of restarts. With this difference in assumptions of the two algorithms, we conducted an empirical comparison using the two sample problems provided along with Harmonica's authors implementation. As shown in Table 14 , the results show that SignHunter optimizes the two problems with $8\times$ less number of queries than Harmonica, not to mention the significant computational advantage.

Table 14: Performance comparison of SignHunter and Harmonica on two sample problems from https://github.com/callowbird/Harmonica. The lower solution quality, the better.

|  | Algorithm | Solution Quality | # queries | Time per Query |
|---|---|---|---|---|
| **Problem 1** | Harmonica | -50.0 | 4223 | 67.47 ms |
|  | SignHunter | -50.0 | 20 | $36.30\mu s$ |
| **Problem 2** | Harmonica | -916.22 | 4223 | 60.22 ms |
|  | SignHunter | -916.21 | 500 | $584.28\mu s$ |

## APPENDIX I. ON THE $\ell_2$-$\ell_\infty$ PERFORMANCE GAP

As discussed in Section 4, in an $\epsilon - \ell_2$ threat setup, black-box attacks that are based on continuous optimization (e.g., NES and $\text{Bandits}_{TD}$) can vary each pixel $x$ within $[x - \epsilon, x + \epsilon]$. On the other hand, SignHunter is restricted to $[x - \epsilon/\sqrt{n}, x + \epsilon/\sqrt{n}]$. In other words, SignHunter in $\epsilon - \ell_2$ perturbation setup behaves exactly the same when used in $\epsilon/\sqrt{n} - \ell_\infty$ perturbation setup. This is illustrated in Figure 14

To highlight the additional perturbation space that other algorithms have over SignHunter in the $\ell_2$ setup, we ran NES and $\text{Bandits}_{TD}$ as representative examples of standard and dimensionality-reduction-based algorithms against the CIFAR10 model used in Section 4 with an $\ell_\infty$ perturbation setup of $\epsilon = 127/\sqrt{n}$. In this and and the $\ell_2$ setup used in Section 4, SignHunter behaves the same, while the performance of NES and $\text{Bandits}_{TD}$ drops significantly from their $\ell_2$ performance due to the reduction in the perturbation space.

A possible fix to allow SignHunter to access the additional search space introduced in the $\ell_2$ setup is to extend the notion of binary sign flips over $\{+1, -1\}$ to ternary sign flips over $\{+1, 0, -1\}$ and we intend to explore this thoroughly in a future work.

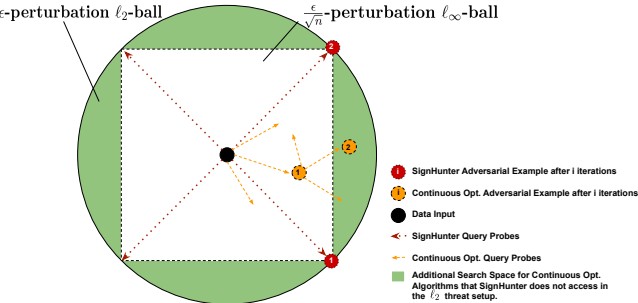

Figure 14: Illustration of adversarial examples crafted by SignHunter in comparison to attacks that are based on the continuous optimization (e.g., NES and $\text{Bandits}_{TD}$) in both (a) $\ell_\infty$ and (b) $\ell_2$ settings. For both $\epsilon$-$\ell_2$ and $\epsilon/\sqrt{n}$-$\ell_\infty$ perturbation balls, SignHunter behaves the same, while continuous attacks such as NES have access to more possible perturbations in the $\ell_2$ setup compared to their perturbations in the $\ell_\infty$ setup. This is demonstrated on CIFAR10 in Figure 15.

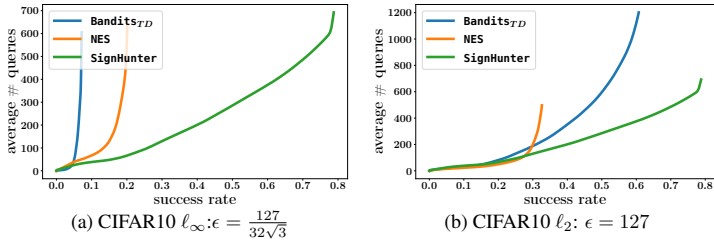

Figure 15: Performance of black-box attacks in the $\ell_\infty$ and $\ell_2$ perturbation constraints. The plots show the average number of queries used per successful image for each attack when reaching a specified success rate. Note that (b) is similar to the $\ell_2$ setup examined in Section 4.

