# OpenReview forum: "Sign Bits Are All You Need for Black-Box Attacks"
_ICLR.cc/2020/Conference — Accept (Poster)_

### Official Review · AnonReviewer3 · 2019-10-21
**Official Blind Review #3**

**Rating:** 6

**Review:**

I'm satisfied with the response. I'll keep my original rating towards acceptance.

----------------------------

This paper proposes a black-box adversarial attack method to improve query efficiency and attack success rate. Instead of estimating the gradient of a black-box model, the proposed method estimates the sign of the gradient, which is an easier task. The SignHunter algorithm is proposed to estimate the sign of the gradient by a divide-and-conquer search. And adversarial examples are generated based on the sign gradient. Extensive experiments prove the effectiveness of the proposed attack method.

Overall, the proposed method on estimating the sign of the gradient for black-box attacks is novel. The authors provide sufficient analysis to present the algorithm, making it clear to see the advantage of the proposed method over previous methods. I have some minor comments on this paper.

1. In Section 2 (Line 15), the authors argue that "However, the queries are not adaptive, they are constructed based on i.i.d. random vectors {vi}". However, in Ilyas et al. (2019), the queries are designed based on the time prior, which are actually adaptive.

2. Estimating the sign of the gradient is more suitable for attacks based on the l_\infty norm, but could affect the results for l_2 attacks, since it reduces the search space. The experiments also show that the proposed method for l_2 attacks is not as effective as l_\infty attacks.

3. It's better to show some tabular results in Section 4 rather than appendix to compare the performance with numbers.

**Experience Assessment:**

I have published one or two papers in this area.

**Review Assessment: Checking Correctness Of Derivations And Theory:**

I carefully checked the derivations and theory.

**Review Assessment: Checking Correctness Of Experiments:**

I carefully checked the experiments.

**Review Assessment: Thoroughness In Paper Reading:**

I read the paper at least twice and used my best judgement in assessing the paper.

---

> ### Author Response · Authors · 2019-11-11
> **Response to Official Blind Review #3**
>
>
> We thank the reviewer for the insightful remarks. Please find the responses below.
>
> 1. We use the term adaptive to characterize how the algorithm constructs its perturbation vector $q_t$ deterministically based on the previous perturbations. Mathematically, the perturbation vector at time $t$ can be expressed as $q_t = \mathbf{s}_t (\mathbf{1}_{keep}q_{t-1}+ (1 -\mathbf{1}_{keep}) q_{t-2})$, where $\mathbf{s}_t$ is the sign flip mask at step $t$ and $\mathbf{1}_{keep}$ is the indicator vector to keep the sign flips at step $t-1$--multiplications are element-wise. In contrast, other algorithms construct I.I.D randomly perturbation vectors: $q_t \sim \mathcal{N}(0,I)$. We have added a footnote that discusses the above (Section 1, Page 2).
>
> 2. The reviewer is correct. We have added a new appendix (Appendix I) that discusses the reduction in the search space for SignHunter in the $\ell_2$ setup in more detail.
>
> 3. Thank you for the remark. As a representative of the datasets, we have included a tabulated summary of the CIFAR10 experiments in Section 4.

---

### Official Review · AnonReviewer1 · 2019-10-22
**Official Blind Review #1**

**Rating:** 6

**Review:**

Summary:
This paper proposes a black-box adversarial sample generation algorithm, which proceeds by learning the signs of the gradient using an adaptive query scheme, which the authors call _SignHunter_. The authors begin by deriving an upper bound on the query complexity for _SignHunter_. Next, the authors empirically compare the performance of _SignHunter_
with existing sign-based optimization schemes in the literature, and demonstrate the advantages of _SignHunter_.


Main Comments:

- Theoretical analysis of the Adaptive Scheme: The authors emphasize the adaptive nature of the proposed scheme as one of the contributions. However, this claim is not justified theoretically:  it would be interesting if the benefits of adaptivity could be quantified in terms of improved worst-case query complexity etc. as compared to any non-adaptive scheme. The query complexity of 2^{\log(n)+1} given in Theorem 1 is not very informative, since a lower complexity (n) is achieved by a simple open-loop  scheme which serially queries each coordinate (i.e., the first query is coordinate 1, the second query is coordinate 2, and so on).

-  Comparison with Hazan el. al (2018): A closely related work in optimizing real-valued functions with domain \{-1,1\}^n is Hazan et. al published in ICLR 2018, which assumes that the black-box function is sparse or compressible in the Fourier domain, and employs compressed sensing techniques to get fast convergence rates in the optimization error. Since that paper considers a very similar optimization problem and proposes a scheme with provable convergence guarantees,  I think it is important that the authors compare,  either theoretically or empirically,  the performance of their proposed scheme with Harmonica (the algorithm of Hazan et al (2018)) to justify the benefits their proposed scheme.

- Parameter Free Algorithm: The statement of Theorem 1 contains the following hypothesis: "the directional derivative is well approximated by the finite difference (Eq. 1)". This condition must be clarified, because it seems to contradict the claim made by the authors later in the last line of Page 5 that _SignHunter_ is "parameter-free" as they set \delta= \epsilon. The condition in the statement of Theorem 1 will not necessarily be satisfied for any \delta, and admissible values of \delta must depend on the properties of the function. Informally, I think that the condition will only be satisfied for \delta "small enough", and one way to make precise the meaning of "small enough" is in terms of the Lipschitz constant of the neural network. However, in that case, I am not sure that the algorithm would remain parameter-free.  I think the authors should make the assumption in Theorem 1 precise and derive suitable sufficient conditions on \delta under which the assumption is satisfied.

References:
1. Hazan, Elad, Adam Klivans, and Yang Yuan. "Hyperparameter optimization: A spectral approach." ICLR (2018).


**Experience Assessment:**

I have read many papers in this area.

**Review Assessment: Checking Correctness Of Derivations And Theory:**

I carefully checked the derivations and theory.

**Review Assessment: Checking Correctness Of Experiments:**

I assessed the sensibility of the experiments.

**Review Assessment: Thoroughness In Paper Reading:**

I read the paper thoroughly.

---

> ### Author Response · Authors · 2019-11-11
> **Response to Official Blind Review #1**
>
> We thank the reviewer for the insightful comments and remarks that improved the paper significantly. Please find our responses below.
>
> - $\textbf{Theoretical analysis of the Adaptive Scheme}$:
> We acknowledge the limitations of the complexity upper bound  while pointing out that no other black-box bound exists. We can clarify our intent. The theorem is not intended to derive or motivate the algorithm, rather we are trying to draw attention to the cost of full sign recovery so we can contrast how  SignHunter succeeds with only partial recovery. We think it is of interest that SignHunter demonstrably outperforms other algorithms with tighter and better bounds because they have to complete all queries (wrt to their bounds) while SignHunter can exit much earlier. To demonstrate this, we ran over all the dataset/perturbation setups the open-loop scheme of sequential sign flips, which as the reviewers pointed out has a better query bound of $O(n)$ in addition to other sign flip schemes. We have put the results in a new appendix ($\textbf{Appendix G}$). The results highlight the advantage of SignHunter’s sign flips over the sequential flips, despite having a worse upper-bound on the query complexity.
>
> - $\textbf{Comparison with Hazan el. al (2018)}$:
> Thank you for drawing this interesting connection. Indeed, both algorithms seek to optimize a black-box function over the binary hypercube $[-1,+1]^n$, albeit with different assumptions on the objective function. As the reviewer pointed out, Harmonica assumes that the objective function can be approximated by a sparse and low degree polynomial in the Fourier basis. Our assumption with SignHunter is that the objective function is separable (Property 1 in Section 3), this lets us optimize the black-box function with $O(n)$ queries given an initial guess instead of searching over the $2^n$ vertices. If this assumption is not met, we can restart SignHunter with another guess with a search complexity of $O(mn)$ where m is the number of restarts. With this difference in assumptions of the two algorithms, we conducted an empirical comparison using two sample problems provided by Harmonica’s authors and used their implementation to run Harmonica. The comparison is illustrated in the newly added notebook in the code’s repo (https://github.com/sign4bb/sign_4_bb/tree/master/harmonica). The results show that SignHunter optimizes the two problems with 8x less number of queries than Harmonica, not to mention its significant computational advantage. The comparison is tabulated and discussed in $\textbf{Appendix H}$ of the revised manuscript.
>
> - $\textbf{Meaning of Parameter-Free}$:
> We view the algorithm as parameter-free viewing $\epsilon$ as a constraint on the allowed perturbation that is imposed upon all attacks and as the algorithm always takes $\delta$ as $\epsilon$ without any tuning.  This is just like it needs to know the dimensionality $n$ of the model’s input. The other algorithms, on the other hand, tune $\delta$ for each dataset/perturbation constraint besides other parameters (see Appendix C, Tables 2 to 6). We revised our use of parameter-free to stress the tuning-free aspect of the algorithm, thank you for the remark.  All black-box attacks that rely on finite-difference approximation, including $\texttt{Signhunter}$, assume the directional derivative is well approximated by the finite-difference. We reiterate this prior to the proof in Appendix B in the revised manuscript.

---

> > ### Comment · AnonReviewer1 · 2019-11-15
> > **reasonable responses**
> >
> > I thank the authors for their clarifications.
> >
> > 1. **Regarding Theoretical Analysis.** Thanks for clarifying the intent behind Theorem 1. Given that _SignHunter_ performs much better in practice than what is implied by the upper bound, it is an interesting question to precisely characterize the subset of separable functions for which _SignHunter_ is much faster. Nevertheless, the additional figures of Appendix~G are helpful as they empirically verify the claim made by the authors that _SignHunter_ can stop only after a fraction of the number of samples that the upper bound implies.
> >
> > 2. **Comparison with Harmonica.** Thanks for carrying out an empirical comparison of _SignHunter_ with Harmonica of Hazan et. al (2017), and demonstrating the lower computational cost as well as better query complexity of _SignHunter_.
> >
> > 3. **Parameter-free assumption.** Thanks for the clarification. It would be helpful to include the references in Appendix-B to prior works which make the assumption that the derivatives are well approximated by finite differences.
> >
> > While the theoretical bound presented in Theorem~1 is not very informative, the authors seem to have empirically established the fact that in practice _SignHunter_ requires much fewer samples to identify a reasonably good gradient direction. Also, the additional results comparing _Signhunter_ with Harmonica in Appendix-H provide further evidence of the strong empirical performance of _Signhunter_. Finally, having gone through the comments of the other reviewers about the novelty and simplicity of the proposed scheme, I will update my score to _Weak Accept (6)_.

---

### Official Review · AnonReviewer2 · 2019-10-22
**Official Blind Review #2**

**Rating:** 8

**Review:**

In this paper, the authors introduce a black box adversarial attack based on estimating the sign of the gradient. To estimate this more efficiently than the gradient itself, the authors exploit the fact that directional derivatives can be estimated using finite differences using only two function evaluations, and use a simple flip/revert procedure to estimate a number of sign bits simultaneously. This sign bit gradient vector is then used in place of the true gradient in an FGSM-like procedure. The central arguments are that (1) estimating the sign bits alone is sufficient to produce adversarial examples, and (2) this can be done quickly enough in practice to yield an efficient procedure.

Overall, I feel that this paper makes a decent contribution to blackbox adversarial generation in settings where confidence scores are available. In particular, many algorithms in this area are often quite complicated and involve machinery like genetic programming. Recent work has begun to demonstrate that significantly simpler routines can not only generate adversarial images, but can do so with significantly fewer queries than their more complicated counterparts.

In particular, two of the methods compared to (NES, Bandits-TD) are state of the art or nearly state of the art, and seem to be significantly outperformed in most regimes considered -- at a glance it appears this approach may outperform other recent work that isn't compared to, such as the method of Guo et al., 2019. The inclusion of results on the public blackbox attack challenges is also welcome.

Could the authors comment on the apparent degradation of performance on L2 performance as the image dimensionality increases? Is this simply an artifact of the fact that the ||x||_{2} <= sqrt(n) ||x||_{\infty} bound directly scales with the input dimensionality? It would be interesting to verify this shortcoming by determining whether applying recent techniques for dimensionality reduction and approximating signed gradients in the subspace alters the relative performance of methods in the L2 perturbation constraint experiments.

**Experience Assessment:**

I have published one or two papers in this area.

**Review Assessment: Checking Correctness Of Derivations And Theory:**

I assessed the sensibility of the derivations and theory.

**Review Assessment: Checking Correctness Of Experiments:**

I carefully checked the experiments.

**Review Assessment: Thoroughness In Paper Reading:**

I read the paper thoroughly.

---

> ### Author Response · Authors · 2019-11-11
> **Response to Official Blind Review #2**
>
> We thank the reviewer for their appreciation of our contribution’s effectiveness and simplicity.  We are pleased s/he  noted our NES, Bandits-TD and public black box attack challenges results.
>
> - $\textbf{Comparison with Recent Works}$
> The comparison to Guo et al. is a bit involved (and also requested by Maksym Andriushchenko in a public comment as item [2]). First, as we have highlighted in the discussion of our results, SignHunter is more suited to the $\ell_\infty$ perturbation setup. Nevertheless, for completeness, our experiments report performance on both $\ell_\infty$ and $\ell_2$ perturbation threats. On the other hand, Guo et al reports just L2 results and against models different from those considered in our paper—except for the IMAGENET v3 model, which the authors find much more difficult to attack. The v3 curves at 10, 000 queries in (Guo et al., 2019, Figure 4) for SIMBA (and its variant SIMBA-DCT) look comparable to SignHunter’s of Figure 9 in our paper. Therefore, one can’t establish a direct comparison without dedicated experiments.
> Second, looking at Guo et al’s algorithm (SIMBA) implementation (https://github.com/cg563/simple-blackbox-attack/blob/master/simba_single.py), it appears that it performs a single random coordinate flip at a time over {+1, 0,-1}. In our work towards SignHunter, we have found out that flipping one coordinate at a time is inefficient without dimensionality reduction.
>
> For the two points above, we have conducted two experiments.  In the first experiment, we implemented Guo et al’s algorithm and evaluated it against the CIFAR10 model in Section 4 for both $\ell_\infty$ and $\ell_2$. Results are discussed in Appendix H and they show that SIMBA is comparable to SignHunter in the $\ell_2$ setup but its performance drops significantly in the $\ell_\infty$ setup.,  In the second experiment, we implement SIMBA’s single coordinate flips in the framework of SignHunter. That is, similar to SIMBA, we flip one coordinate at a time but the flips are over {-1,+1} instead of Guo et al.’s {+1,0,-1}, we refer to this scheme as Naive. Results are shown in Appendix G and they demonstrate that single coordinate flips are not as effective as SignHunter’s adaptive coordinate flips.
>
> In addition to Guo et al’s work, we have discussed two other related recent works in the rest of Appendix G.
>
> - $\ell_2-\ell_\infty \textbf{Performance Gap}$:
> We believe the relative performance degradation form Linf to L2 is primarily due to the reduced perturbation space that SignHunter has compared to other algorithms — it only deals with {+1,-1} perturbations per pixel whereas other algorithms deal with {+1, 0, -1}.  As discussed in Section 4 ($\ell_\infty$ vs. $\ell_2$ Perturbation Threat.), in an $\epsilon \ell_2$ threat setup, other algorithms can vary each pixel x within $[x-\epsilon,x+\epsilon]$. On the other hand, SignHunter is restricted to $[x-\epsilon/\sqrt{n}, x+\epsilon/\sqrt{n}]$. In other words, SignHunter in $\epsilon \ell_2$ perturbation setup behaves exactly the same when used in $\epsilon/\sqrt{n} \ell_\infty$ perturbation setup.
> We  add a new experiment in a new appendix (Appendix I) to intentionally highlight the additional perturbation space that other algorithms have over SignHunter in the $\ell_2$ setup by choosing to compare against NES and Bandits-TD as representative examples of standard and dimensionality-reduction-based algorithms against the CIFAR10 model used in Section 4 with an $\ell_\infty$ perturbation setup of $\epsilon=127/\sqrt{n}$. In this and and the $\ell_2$ setup used in Section 4, SignHunter behaves the same, while the performance of NES and Bandits-TD drops significantly from their $\ell_2$ performance due to the reduction in the perturbation space.
>
> A simple fix to this $\ell_\infty-\ell_2$ performance gap would be to extend the notion of binary sign flips over {+1,-1} to ternary sign flips and we intend to explore this thoroughly in future work.

---

### Public Comment · ~Maksym_Andriushchenko1 · 2019-11-05
**Comparison to the prior work**

Thank you for the interesting paper.

I'm quite curious about a comparison to the following two ICML'19 papers:
[1] Parsimonious Black-Box Adversarial Attacks via Efficient Combinatorial Optimization, ICML'19, https://arxiv.org/abs/1905.06635
[2] Simple Black-box Adversarial Attacks, ICML'19, https://arxiv.org/abs/1905.07121

Note that these two papers are state-of-the-art in terms of query-efficiency for Linf/L2 norms respectively (at least among published papers), and they significantly outperform the Bandits approach. Thus, I think the paper would benefit from comparison to these methods as well.

---

> ### Author Response · Authors · 2019-11-11
> **On Recent Related Work**
>
> We thank you for your interest.
>
> With regard to [1], our experiment on the public CIFAR10 black-box attack challenge corresponds to [1, Table 1]. The authors report a 48% success rate (52% model accuracy) with an average number of queries of 1261. On the other hand, our proposed algorithm achieves a 52.84%  success rate (47.16% model accuracy) with an average number of queries of 569. Further, [1, Table 2] corresponds to our Appendix D, Table 9; where the authors report a 98.5% success rate with an average number of queries of 722. Our proposed algorithm achieves a 98% success rate with 578.56 average number of queries. Based on these numbers, SignHunter demonstrates better performance than [1]’s attack.
>
> Please see how we have addressed [2] in our response to Blind Review #2, above.
>
> We have incorporated the discussion of the above and other related work in Appendix H of the revised manuscript.

---

### Decision · Program_Chairs · 2019-12-19

**Decision:**

Accept (Poster)

**Comment:**

This paper presents a novel black-box adversarial attack algorithm, which exploits a sign-based rather than magnitude-based, gradient estimator for black-box optimization. It also adaptively constructs queries to estimate the gradient. The proposed approach outperforms many state-of-the-art black-box attack methods in terms of  query complexity. There is a unanimous agreement to accept this paper.